# Elevated pre-mRNA 3′ end processing activity in cancer cells renders vulnerability to inhibition of cleavage and polyadenylation

Yange Cui [1,4], Luyang Wang[1,4], Qingbao Ding[1], Jihae Shin[2], Joel Cassel[3], Qin Liu [3], Joseph M. Salvino [3] & Bin Tian [1] ✉

Cleavage and polyadenylation (CPA) is responsible for 3′ end processing of eukaryotic poly(A)+ RNAs and preludes transcriptional termination. JTE-607, which targets CPSF-73, is the first known CPA inhibitor (CPAi) in mammalian cells. Here we show that JTE-607 perturbs gene expression through both transcriptional readthrough and alternative polyadenylation (APA). Sensitive genes are associated with features similar to those previously identified for PCF11 knockdown, underscoring a unified transcriptomic signature of CPAi. The degree of inhibition of an APA site by JTE-607 correlates with its usage level and, consistently, cells with elevated CPA activities, such as those with induced overexpression of FIP1, display greater transcriptomic disturbances when treated with JTE-607. Moreover, JTE-607 causes S phase crisis and is hence synergistic with inhibitors of DNA damage repair pathways. Together, our data reveal CPA activity and proliferation rate as determinants of CPAi-mediated cell death, raising the possibility of using CPAi as an adjunct therapy to suppress certain cancers.

Almost all eukaryotic protein-coding and long noncoding genes employ cleavage and polyadenylation (CPA) for 3′ end processing of their pre-RNA[1]. The site for CPA, also known as the polyadenylation site (PAS), is defined by surrounding sequence motifs[2,3]. The constellation of PAS motifs determine the strength, or usage level, of a given PAS[4]. Most mammalian genes have multiple PASs, resulting in expression of alternative cleavage and polyadenylation (APA) isoforms[5–9]. For mRNA genes, APA in the last exon generally alters the length of 3′ untranslated region (3′UTR)[10], whereas those in introns lead to transcripts with different coding sequences or truncated, unstable transcripts[11–13]. The APA site usage profile can vary substantially among cell types[14,15], and during cell differentiation and development[16,17]. Owing to their high proliferative rates, cancer cells generally favor proximal APA site usage[18]. However, considerable APA isoform expression differences have been reported in different cancer types[19].

The PAS is recognized by the CPA machinery, which is composed of over 20 core factors in mammalian cells[1,20,21]. Most of the CPA proteins form distinct sub-complexes, including CPSF, CstF, CFI and CFII. CPSF can be further divided into two functional modules[20], namely, mammalian polyadenylation specificity factor (mPSF), including CPSF-160, CPSF-30, WDR33, and FIP1, and mammalian cleavage factor (mCF), including CPSF-100, CPSF-73, and Symplekin. In addition, CFI includes CFI-25, CFI-59, and CFI-68; CstF includes CstF-50, CstF-64 and CstF-77; CFII includes CLP1 and PCF11. Moreover, RBBP6, Poly(A) polymerase, PAPN1, and RNA polymerase II (Pol II) are also key components of the CPA machinery[1]. Among the growing number of mechanisms that impact APA isoform expression, regulation of core factor expression level appears to have a substantial influence on APA site usage, resulting in global shifts between proximal and distal PAS isoforms[7]. This regulatory scheme has been implicated in 3′UTR size regulation in cell proliferation and differentiation[17,22].

[1]Gene Expression and Regulation Program, and Center for Systems and Computational Biology, The Wistar Institute, Philadelphia, PA 19104, USA. [2]Department of Microbiology, Biochemistry and Molecular Genetics, Rutgers New Jersey Medical School, Newark, NJ 07103, USA. [3]Molecular and Cellular Oncogenesis Program, The Wistar Institute, Philadelphia, PA 19104, USA. [4]These authors contributed equally: Yange Cui, Luyang Wang. ✉e-mail: btian@wistar.org

In addition to 3′ end processing, CPA plays a critical role in termination of transcription[23], which takes place within a variable region after the PAS[24]. Inhibition of CPA, here termed CPAi for simplicity, could lead to transcriptional readthrough[25–27], a phenomenon that has been observed in cells under stress conditions[28,29] and upon viral infection[30–32], as well as in certain cancer cells[33]. Transcriptional readthrough could cause transcriptional interference between neighboring genes[23] and lead to chimeric transcripts containing sequences from two adjacent genes[33].

Developed about 20 years ago as an inhibitor for cytokine production in human cells[34], JTE-607 is a compound which was recently found to inhibit the activity of CPSF-73[35,36], the endonuclease in CPA machinery. While CPSF-73 inhibitors have been used in multiple parasitic protozoans[37,38], JTE-607 is the first known CPAi compound in human cells. JTE-607 has also been shown to suppress certain cancer cells, including those from acute myeloid leukemia[39], Ewing's sarcoma[36], some lung and hepatocellular carcinoma[36], breast cancer[40] and pancreatic cancer[41]. However, some cancer cells appear quite tolerant of the compound[36], raising the question as to what cellular features are deterministic for cell survival upon JTE-607 treatment. Here, we profile mature poly(A)+ RNA and nascent, chromatin-bound RNA in multiple cancer cells treated with JTE-607. We identify genomic and RNA features associated with gene expression disturbance by JTE-607. By analyzing APA regulation, we reveal rules by which JTE-607 inhibits PAS usage. We show that the CPA activity of a cell is a key determinant of its responsiveness to JTE-607. In addition, cell proliferation contributes to JTE-607-elicited cell death, due likely to S phase crisis and DNA damage caused by transcription-replication conflicts. We further examine synergy between JTE-607 and DNA damage repair-based inhibitors. Our results raise the prospect of CPAi-based therapeutics for suppressing certain cancers.

## Results

### Widespread gene expression changes in HeLa and HepG2 cells treated with JTE-607

We were interested in understanding genomic and cellular features associated with cell sensitivity to JTE-607, the first known CPAi compound for mammalian cells[36]. To this end, we first used human cervical carcinoma cell line HeLa and the hepatocellular carcinoma cell line HepG2 (Fig. 1a) because of their markedly different tolerance levels to JTE-607 (IC$_{50}$ > 50 μM for HeLa and <10 μM for HepG2). We subjected these two cell lines to 1 and 10 μM of JTE-607 for 8 h, and used the QuantSeq sequencing method to examine poly(A)+ RNAs (See Methods). Note that QuantSeq reads are biased to the 3′ end, enabling analyses of both gene expression and APA[42].

We observed widespread gene expression changes ($P < 0.05$ after BenjaminiHochberg [BH] adjustment for false discovery rate, Fisher's exact test; fold change ≥ 1.2; Fig. 1b, c) in both HeLa and HepG2 cells after JTE-607 treatment. In each cell line, as expected, 10 μM JTE-607 led to more genes regulated as compared to 1 μM JTE-607 (Fig. 1b, c). Consistent with the IC$_{50}$ difference, HepG2 cells displayed more regulated genes than HeLa cells did at either JTE-607 concentration (Fig. 1b, c). Interestingly, there were more genes

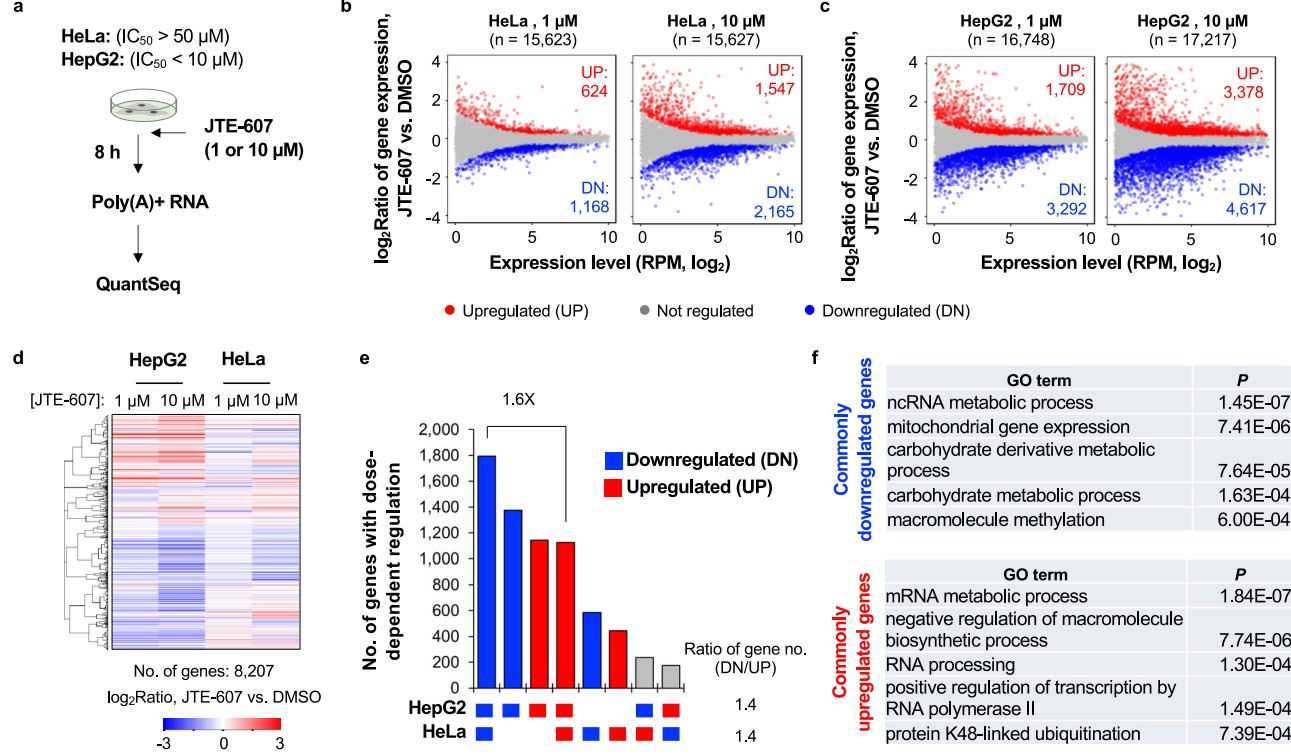

**Fig. 1 | Widespread gene expression changes in HeLa and HepG2 cells treated with JTE-607. a** Schematic showing JTE-607 treatments to HeLa and HepG2 cells, followed by mRNA analysis by using the QuantSeq method. **b** Scatter plots showing gene expression level change (y-axis) vs. expression level (x-axis) after 1 μM (left) or 10 μM (right) JTE-607 treatment in HeLa cells. Each dot is a gene. Significantly regulated ones (BH-adjusted $P < 0.05$, Fisher's exact test; fold change >1.2) are highlighted in red (upregulation, UP) or blue (downregulation, DN). The numbers of UP and DN genes are also indicated. **c** As in **b** except that data are for HepG2 cells. **d** Heatmap showing commonly expressed genes in HepG2 and HeLa cells that were significantly regulated in at least one sample after JTE-607 treatment. Genes were clustered based on expression changes (log$_2$Ratio, JTE-607 vs. DMSO). The number of genes in the heatmap is indicated. **e** UpSet plot summarizing data in **d**. The number of genes with dose-dependent regulation after JTE-607 treatment in HepG2 or HeLa cells are shown. Genes are divided into eight groups based on their regulation in the two cell lines. Gray bars indicate different directions of regulation in the two cell lines. The ratio of downregulated gene number to upregulated gene number (DN/UP) is indicated. **f** Gene ontology (GO) terms associated with genes downregulated (top) or upregulated (bottom) in both HeLa and HepG2 cells after JTE-607 treatment (indicated in **d**). Source data are provided as a Source Data file.

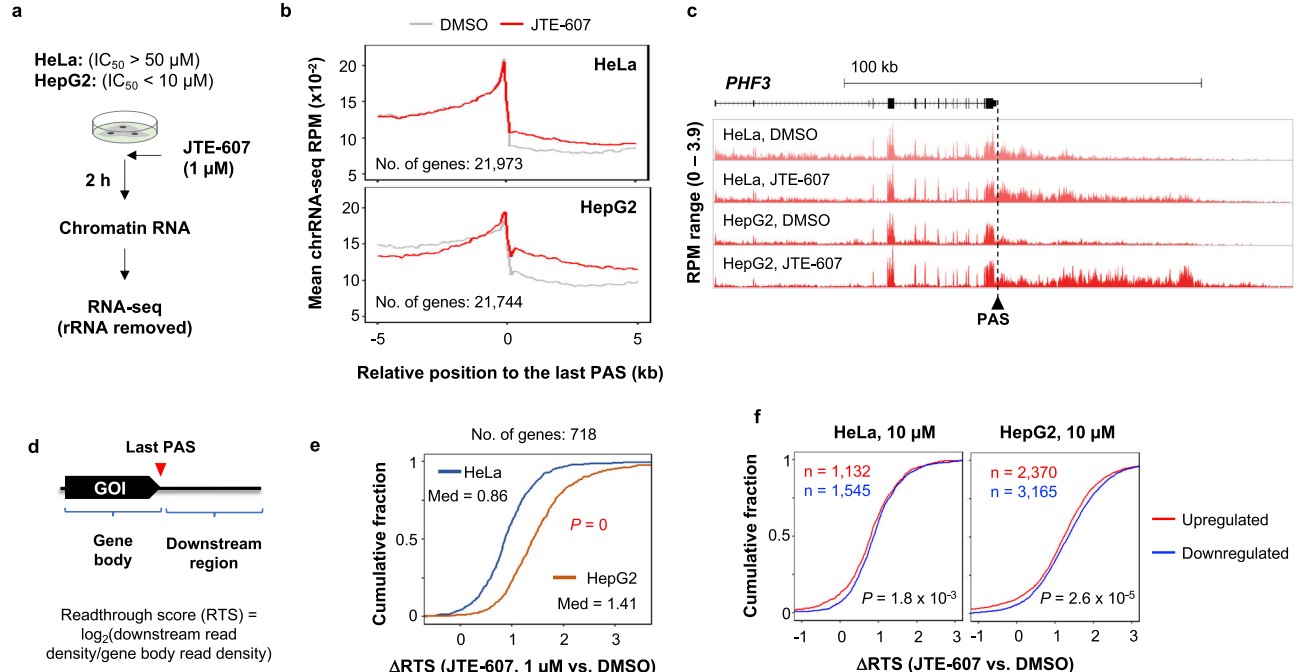

**Fig. 2 | JTE-607 elicits widespread transcriptional readthrough. a** Schematic showing JTE-607 treatments to HeLa and HepG2 cells, followed by chromatin RNA extraction and sequencing. **b** Metagene plots of chrRNA-seq reads in the genomic region around the last annotated PAS (±5 kb) in HeLa (top) or HepG2 (bottom) cells after JTE-607 (red line) or DMSO (gray line) treatment. The chrRNA-seq reads were normalized for each gene. The number of genes used in each plot is indicated. **c** An example gene *PHF3* showing transcriptional readthrough after JTE-607 treatment as indicated by the chrRNA-seq read profiles. **d** Schematic showing calculation of readthrough score (RTS) for gene of interest (GOI). RTS is ratio (log$_2$) of chrRNA-seq read density in the 4 kb downstream region to that of gene body (defined as the region between the transcriptional start site and the last PAS). **e** Cumulative distribution function (CDF) curves of RTS difference (ΔRTS) between JTE-607-treated and DMSO-treated samples for HepG2 (orange line) and HeLa (blue line) cells. The number of genes analyzed, median (med) for each cell line, and *P* (K–S test) comparing ΔRTS between the two cell lines are indicated. **f** CDF curves of ΔRTS for upregulated (red) and downregulated (blue) genes in HeLa (left) and HepG2 (right) cells after 10 μM JTE-607 treatment (Fig. 1b, c). Number of genes for each group is indicated. *P* values (K–S test) indicating significance of ΔRTS difference between upregulated and downregulated genes are shown. Source data are provided as a Source Data file.

downregulated than upregulated (note the respective gene numbers in Fig. 1b, c), a trend that became more pronounced when a fold change ≥2 was used to select genes (Supplementary Fig. 1a). However, it should be noted that our gene expression analysis was based on the assumption that the total amount of poly(A)+ RNA was constant before and after treatment, which may not hold in all conditions.

Pair-wise scatter plot analysis (Supplementary Fig. 1b) and cluster analysis (Fig. 1d) of genes that were commonly expressed in both HeLa and HepG2 cells showed general correlations in gene expression regulation between these two cell lines, indicating that JTE-607 impacted gene expression similarly in HepG2 and HeLa cells, despite that the degree of regulation was greater in the former than the latter. These results were further corroborated by the UpSet plot analysis using only genes showing dose-dependent regulation (stronger regulation after 10 μM treatment than 1 μM, Fig. 1e): more genes were regulated in HepG2 cells than HeLa cells, and downregulated genes in both cell lines were 1.6-fold greater in number than upregulated ones. Gene ontology (GO) analysis showed distinct functions associated with different gene groups in the UpSet plot (Fig. 1f and Supplementary Fig. 1c). The top GO terms associated with commonly downregulated genes in HeLa and HepG2 cells included 'ncRNA metabolic process', 'mitochondrial gene expression', 'carbohydrate derived metabolic process', etc. (Fig. 1f, top), whereas the top ones associated with commonly upregulated genes included 'mRNA metabolic process', 'negative regulation of macromolecule biosynthetic process', 'RNA processing', etc. (Fig. 1f, bottom). Overall, the GO analysis appeared to indicate that JTE-607 has a pleiotropic effect on gene expression in these two cell lines.

## JTE-607 elicits global transcriptional readthrough

Previous studies of JTE-607 treatment and CPSF-73 knockdown showed transcriptional readthrough in these conditions[25,26,36]. We next carried out chromatin RNA sequencing (chrRNA-seq) to examine transcriptional readthrough, using HepG2 or HeLa cells treated with 1 μM JTE-607. We reasoned that the transcriptional readthrough would be quickly observable after JTE-607 treatment and hence harvested cells after 2 h of treatment (Fig. 2a). Indeed, both cell lines displayed substantial transcriptional readthrough, as indicated by increased chrRNA-seq reads mapped to the region downstream of the last PAS (5 kilobase, kb) across genes (Fig. 2b and Supplementary Fig. 2a). Metagene plots indicated that the transcriptional readthrough appeared stronger in HepG2 cells than in HeLa cells (Fig. 2b and Supplementary Fig. 2a), as exemplified by the gene *PHF3* (Fig. 2c). Using the ratio (log$_2$) of chrRNA-seq read density in the 4 kb downstream region to that in the gene body (from transcriptional start site to the last PAS), we calculated a readthrough score (RTS) for each gene (illustrated in Fig. 2d). Based on difference of RTS between JTE-607-treated and DMSO-treated cells, or ΔRTS, we found that while both HeLa and HepG2 cells displayed positive ΔRTS values (median = 0.86 and 1.41, respectively, Fig. 2e), transcriptional readthrough in HepG2 was significantly greater than HeLa (*P* = 0, K–S test, Fig. 2e). Therefore, the difference in extent of transcriptional readthrough elicited by JTE-607 in HeLa vs. HepG2 cells correlates with gene expression change difference between the two cell lines.

Previous studies have identified genes that tend to have transcriptional readthrough under stress conditions, named Pan-Downstream of Genes (Pan-DoGs)[29]. However, there was no significant difference in ΔRTS values between Pan-DoGs and other genes in our data (HepG2 cells) when gene expression levels were controlled

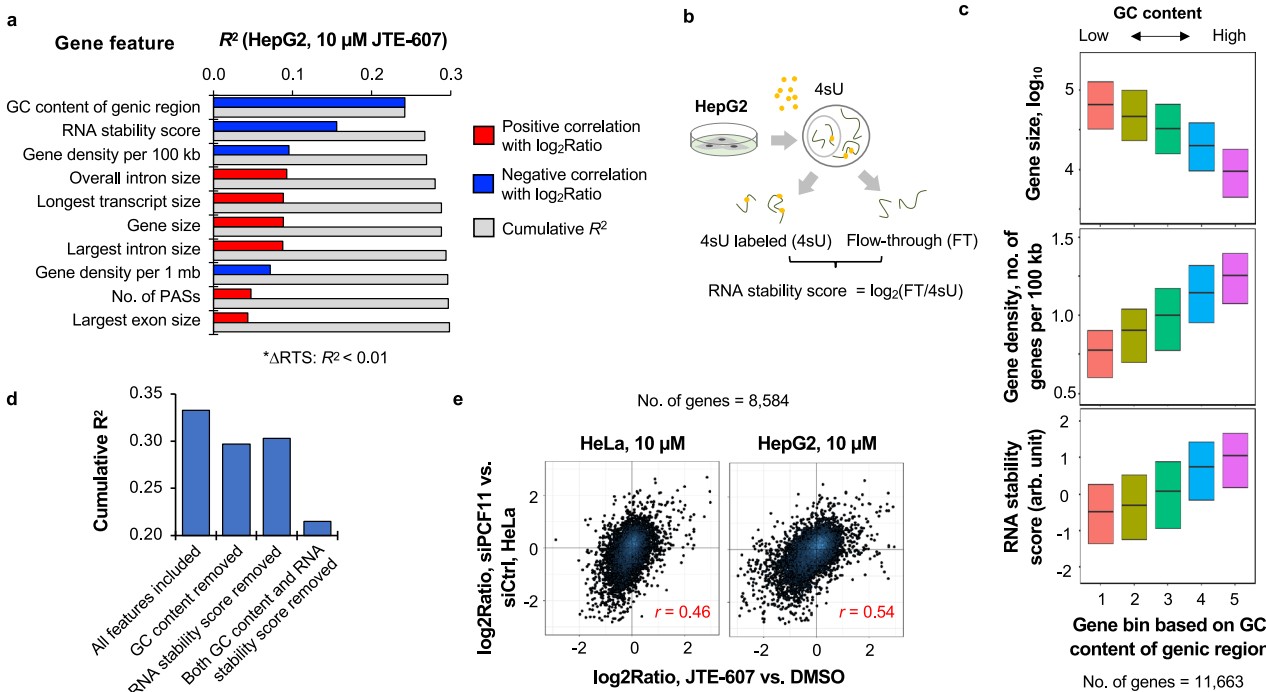

**Fig. 3 | Genomic and RNA features associated with JTE-607-induced gene expression changes. a** Top features correlated with gene expression regulation by JTE-607 (10 μM) in HepG2 cells. Features are sorted by their individual $R^2$ values, with blue color indicating negative correlation with gene expression changes and red color positive correlation. Cumulative $R^2$ of a feature (shown in gray) is based on the feature combined with all other features that have a higher individual $R^2$. **b** Schematic showing calculation of the RNA stability score, which is the ratio ($\log_2$) of RNA abundance of pre-existing RNA (FT) to newly made RNA (4sU labeled). **c** Boxplots showing that GC content is correlated with gene size, gene density, or RNA stability score. All genes with detectable expression ($n = 11,663$) were divided into five equally sized bins based on the GC content of genic region. The top and bottom of each box represent the 75th and 25th percentile values, respectively; the median is shown as a line in the box. **d** Bar graph showing overall $R^2$ after removal of either GC content or RNA stability score feature, or both. **e** Scatter plot showing correlation of gene expression changes ($\log_2$Ratio) between HeLa cells (left) or HepG2 cells (right) treated with 10 μM JTE-607 (this study) and HeLa cells with PCF11 siRNA knockdown (a previous study by Kamieniarz-Gdula et al.). Genes with detectable expression in all samples ($n = 8584$) were used. Pearson correlation coefficient ($r$) is indicated in each plot. Source data are provided as a Source Data file.

(Supplementary Fig. 2b), indicating that, unlike stress-induced transcriptional readthrough, JTE-607-elicited readthrough is quite universal to all genes. In both HeLa and HepG2 cells, downregulated genes had significantly higher ΔRTS values than upregulated genes ($P < 0.01$, K−S test, Fig. 2f), suggesting that transcriptional readthrough could negatively impact gene expression.

### Genomic feature analysis connects JTE-607 treatment with PCF11 inhibition

To gain more insights into what gene features are associated with mRNA expression regulation, we carried out a linear regression analysis comparing mRNA expression changes with various gene features, including GC content, gene size, gene density, etc. (See Methods). We focused on the data from HepG2 cells treated with 10 μM JTE-607 because of its clearer effects on gene expression than other samples (Fig. 3a and Supplementary Fig. 3a). ΔRTS was also used as a feature, with the goal of gauging the contribution of transcriptional readthrough to gene expression changes as compared to other features. In addition, we included a feature related to RNA stability, named RNA stability score (illustrated in Fig. 3b), a value we recently generated based on the ratio of pre-existing RNA abundance to newly made RNA abundance in HepG2 cells by using metabolic labeling with 4-thiouridine[13]. A high RNA stability score thus indicates greater RNA stability.

The cumulative $R^2$ based on all features was 0.33 (Supplementary Fig. 3a), indicating that all features combined could explain 33% of gene expression changes. The top ten features accounted for an $R^2$ value of 0.30 (Fig. 3a). GC content of genic region and RNA stability score were the top two features with the highest individual $R^2$ values

($R^2 = 0.24$ and 0.16, respectively, Fig. 3a), both of which were negatively correlated with gene expression changes ($\log_2$Ratio, JTE-607 vs. DMSO). The next set of features with high $R^2$ values fell into two groups: (1) features related to gene density, which had negative correlations with $\log_2$Ratio, such as gene density per 100 kb or 1 million base region; (2) features related to gene size, which had positive correlations with $\log_2$Ratio, such as overall intron size (all introns combined), gene size, and the size of largest intron.

Some of the top features were correlated to one another. For example, GC content was positively correlated with RNA stability score and gene density, but negatively correlated with gene size (Fig. 3c), raising the possibility that some of the features were confounding. However, removal of either GC content or RNA stability led to an ~10% reduction of overall $R^2$ (Fig. 3d), whereas removal of both decreased the overall $R^2$ by 33% (Fig. 3d), indicating that they each had important contributions to gene regulation despite that they are correlated with each other.

The gene features associated with JTE-607-elicited gene regulation were reminiscent of those we previously identified for gene regulation in mouse myoblast cells after knockdown (KD) of another core CPA factor Pcf11[43]. To explore this further, we used the RNA-seq data for HeLa cells treated with PCF11 siRNAs (siPCF11) that were previously generated by Kamieniarz-Gdula et al.[27]. Interestingly, the gene expression changes in JTE-607-treated HepG2 or HeLa cells were well correlated with those in siPCF11-treated cells ($r = 0.54$ and 0.46, respectively, Pearson correlation, Fig. 3e). Consistently, we found that the top features associated with gene regulation by siPCF11 matched well with those associated with JTE-607-elicited gene regulation (Supplementary Fig. 3b). Together, these data indicate that JTE-607

could phenocopy *PCF11* KD in gene regulation, underscoring a unified transcriptomic signature of CPAi.

### Neighboring gene analysis reveals impacts of transcriptional readthrough on gene expression

We found that ΔRTS had a low correlation value with log$_2$Ratio ($R^2 < 0.01$, Supplementary Fig. 3a), despite an overall negative impact on gene expression (Fig. 2f), suggesting that transcriptional readthrough per se does not downregulate gene expression. Because gene density negatively correlated with log$_2$Ratio, we reasoned that transcriptional readthrough may have differential impacts on genes in different genomic regions, which was also observed with *PCF11* KD[27,43]. To this end, we focused on gene pairs with opposite transcriptional directions, or a tail-to-tail relationship (tail being the end of gene, illustrated in Fig. 4a). We put all gene pairs, i.e., a gene of interest (GOI) and its nearest neighbor gene (NNG), into five equally sized bins with increasing intervening distances (bins 1 and 5 having the shortest and longest distances, respectively). Interestingly, a strong positive correlation between gene pair distance and gene expression change could be discerned in all samples (Fig. 4b), with the data from HepG2 cells treated with 10 μM JTE-607 showing the strongest trend ($P = 0$ for bin 1 genes vs. bin 5 genes, Wilcoxon test, Fig. 4b). This result suggests that some gene downregulation events elicited by JTE-607 could be due to collision of readthrough Pol II traveling on opposite strands.

To address Pol II collision more explicitly, we developed a score, named ChrRNA-seq Overlapping Signal Score (CROSS), based on chrRNA-seq read signals (used as proxies of Pol II abundances) on antisense vs. sense strands in the last 1 kb region of GOI (Fig. 4a). As expected, genes with a shorter distance to its neighbor (tail-to-tail only) displayed higher CROSS values than those with a longer distance in both HeLa and HepG2 cells (Fig. 4c). Importantly, downregulated genes had significantly higher CROSS values than unchanged genes ($P = 1.8 \times 10^{-15}$, K–S test, Fig. 4d). When we separated genes by both CROSS values and the distance between neighboring genes, the genes with higher CROSS values showed more downregulation in all distance groups (Fig. 4e), supporting the notion that Pol II collision, not gene-to-gene distance per se, is chiefly responsible for gene downregulation (summarized in a schematic shown in Fig. 4f).

We next examined gene pairs in the same transcriptional direction, or a tail-to-head relationship (head being the start of gene), with GOI being downstream of NNG (Fig. 4g). While we also saw a positive correlation between gene expression change and gene-to-gene distance, the trend was much weaker compared to tail-to-tail gene pairs (Fig. 4h vs. 4b). Interestingly, when we stratified short-spaced gene pairs (bins 1 and 2) based on expression levels of GOI and NNG, we found that a gene actually tended to be upregulated when its expression level was low (bottom row in Fig. 4i), suggesting that readthrough Pol II from an upstream gene could cause upregulation of the downstream gene, or Pol II read-in (illustrated in Fig. 4j).

To further examine the Pol II read-in scenario, we developed a score named ChrRNA-seq Intervening Signal Score (CRISS), based on change of chrRNA-seq read abundance between two neighboring genes after JTE-607 treatment (Fig. 4g). We found that for genes that were lowly expressed and upregulated by JTE-607, their CRISS values were significantly higher than nonregulated genes ($P = 2.0 \times 10^{-12}$, K–S test, Fig. 4k, left), a trend that was substantially subdued when all genes were used ($P = 7.4 \times 10^{-4}$, K–S test, Fig. 4k, right). This result is in good agreement with the read-in model (Fig. 4j), in which readthrough Pol II from an upstream gene upregulates expression of its downstream gene, especially when the latter is lowly expressed.

### JTE-607 causes widespread APA changes

We next reasoned that by analyzing APA isoform changes, we could gain insights into how JTE-607 regulates PAS usage. To this end, we first examined isoforms using 3′UTR APA sites (last exon only) in HeLa

and HepG2 cells (illustrated in Fig. 5a). We found that most genes in HeLa and HepG2 cells showed 3′UTR lengthening, i.e., increased relative abundance of distal PAS (dPAS) isoform to that of proximal PAS (pPAS) isoform (Fig. 5b). As with gene expression changes, 3′UTR lengthening was stronger in HepG2 cells than HeLa cells (indicated by colors in Fig. 5b), despite general correlations between these two cell lines ($r = 0.60$ for cells treated with 10 μM JTE-607, Pearson Correlation, Supplementary Fig. 4a). Based on dose-dependent APA regulation events, we found that genes displaying 3′UTR lengthening outnumbered those with the opposite trend by 5.3-fold (Fig. 5c). The example gene *SPTLC3* showed dose-dependent 3′UTR lengthening in both HeLa and HepG2 cells (Fig. 5d), with the latter being more obvious than the former (Fig. 5d). Note that the 3′UTR APA elicited by JTE-607 in HeLa and HepG2 cells were also correlated with those in HeLa cells with *PCF11* KD (Supplementary Fig. 4a), highlighting a common 3′UTR lengthening signature induced by CPAi.

We found that genes showing 3′UTR lengthening had a modest, albeit significant, trend of being upregulated at the level of gene expression as compared to those showing 3′UTR shortening ($P = 0.04$, Wilcoxon test, Fig. 5e, left). However, these two gene groups were not significantly different in transcriptional readthrough changes ($P = 0.9$, Wilcoxon test, Fig. 5e, right). Therefore, APA regulation in the last exon by JTE-607 has a mild effect on gene expression and is largely independent of transcriptional readthrough.

We next examined intronic polyadenylation (IPA) regulation by JTE-607, using 3′ terminal (or last) exon PASs (TPAs) as a reference (Fig. 5f). In both HepG2 and HeLa cells, IPA was generally suppressed by JTE-607 treatment (Fig. 5g). Genes showing dose-dependent IPA suppression in both HepG2 and HeLa cells outnumbered those showing the opposite trend by 4.8-fold (Fig. 5h). Interestingly, the genes showing IPA activation were significantly more downregulated than the genes showing IPA suppression ($P < 1.0 \times 10^{-28}$, Wilcoxon test, Fig. 5i, left). In contrast, these two gene groups did not differ in transcriptional readthrough ($P = 0.13$, Wilcoxon test, Fig. 5i, right). The effect of IPA regulation on gene expression is due likely to the fact that IPA isoforms are generally unstable compared to last exon PAS isoforms[13]. An example case is IPA suppression of *PCF11* after JTE-607 treatment, which is coupled with upregulation of the isoforms using last exon PASs (Fig. 5j). As with 3′UTR APA, IPA change correlations could be discerned between JTE-607-treated HepG2 and HeLa cells ($r = 0.57$ for cells treated with 10 μM JTE-607, Pearson Correlation, Supplementary Fig. 4b) as well as between JTE-607 treatment and *PCF11* KD ($r \geq 0.40$ for HepG2 cells treated with JTE-607, Pearson Correlation, Supplementary Fig. 4b), adding IPA suppression as another signature to CPAi-mediated APA regulation.

We noticed that some IPA activation events were caused by downregulation of transcripts using last exon PASs and IPA isoform expression was largely unchanged (an example gene *MRPL12* is shown in Fig. 5k). This pattern appeared to be more obvious for genes with a single PAS in the last exon. Indeed, when we directly compared genes with multiple PASs in the last exon (each with >10% of all isoform expression) and genes with only one major PAS in the last exon (>95% of all isoform expression), we found that the latter were significantly more downregulated than the former after JTE-607 treatment ($P = 0$, K–S test, Supplementary Fig. 4c). This finding is in line with our gene feature analysis result, in which the number of PASs of a gene positively correlated with gene expression changes after JTE-607 treatment (Fig. 3a and Supplementary Fig. 3a). Taken together, our APA analysis results indicate that JTE-607 generally suppresses pPAS usage, the extent of which is greater in HepG2 cells than in HeLa cells. While 3′ UTR APA regulation has only a modest effect on gene expression, IPA regulation significantly impacts gene expression. In addition, genes harboring multiple PASs are less likely to be downregulated after JTE-607 treatment, suggesting that APA sites in the last exon could help mitigate CPAi's effect on gene expression.

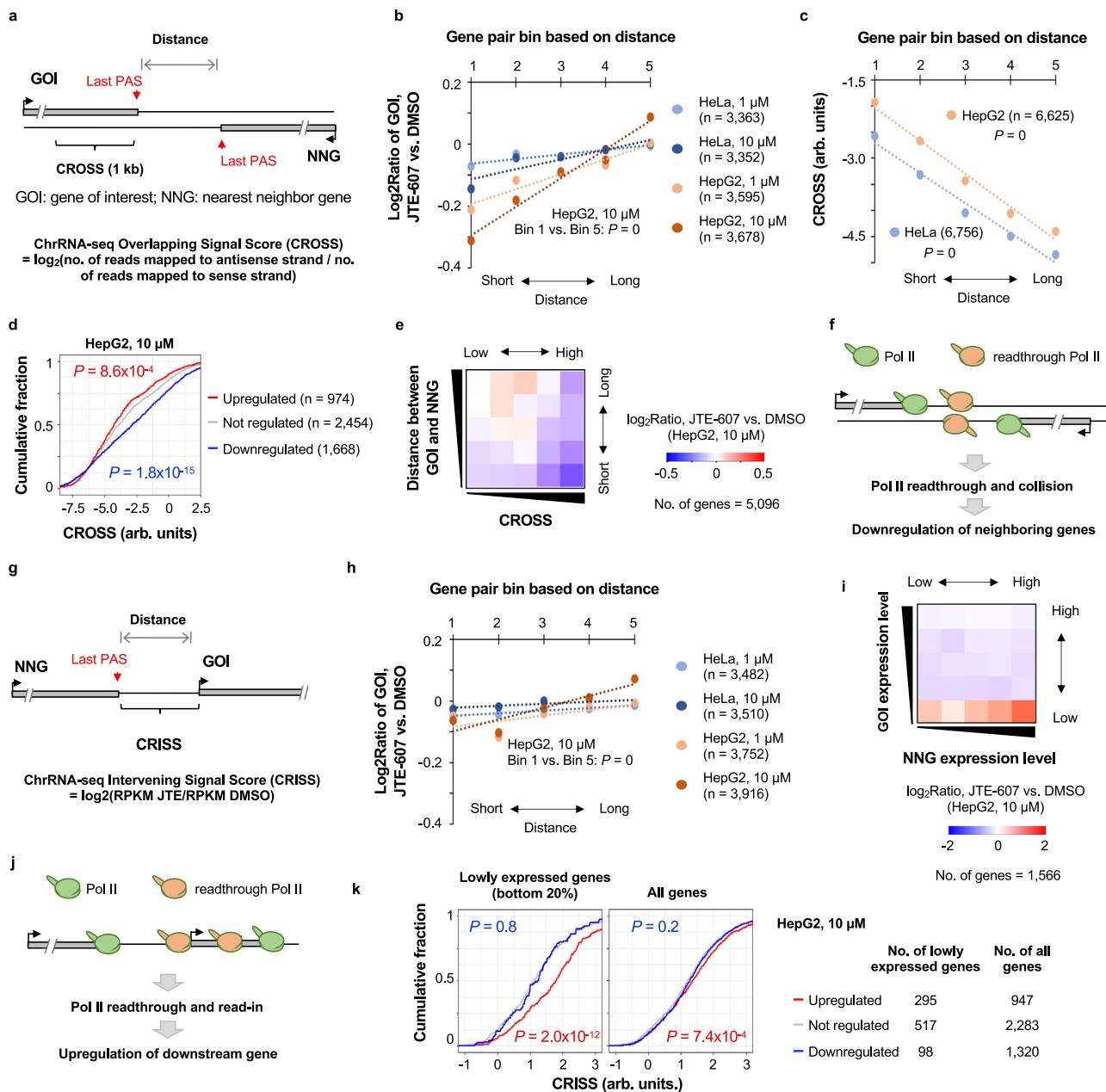

**Fig. 4 | Transcriptional readthrough impacts expression of neighboring genes.** **a** Schematic of a gene pair with opposite transcriptional directions. GOI, gene of interest; NNG, nearest neighbor gene. The last 1 kb region of GOI was used to calculate ChrRNA-seq Overlapping Signal Score (CROSS) as described. **b** Expression changes of GOI as a function of distance from NNG. All GOI-NNG pairs were divided into five equally sized bins based on their distance. The number of genes used in each plot is indicated. *P* value (K−S test) for bin 1 vs. bin 5 comparison is indicated for HepG2 cells treated with 10 μM JTE-607. **c** CROSS as a function of distance between GOI and NNG. *P* value (K−S test) for bin 1 vs. bin 5 comparison is indicated for both HepG2 and HeLa cell data. **d** CDF curves of CROSS for genes regulated in HepG2 cells treated with 10 μM JTE-607. *P* values (K−S test) for comparisons between upregulated genes (red line) or downregulated genes (blue line) and not regulated genes (gray line) are indicated. **e** Heatmap showing gene expression changes based on CROSS (x-axis) and GOI-NNG distance (y-axis). Each gene was assigned to one of the 25 bins based on CROSS and distance values. The median of gene expression changes (log$_2$Ratio in HepG2 cells treated with 10 μM JTE-607) of each bin is represented by color. **f** Schematic indicating that collision of Pol II on two opposite strands leads to gene downregulation. **g** Schematic showing GOI and its upstream NNG having the same transcriptional direction. Calculation of ChrRNA-seq Intervening Signal Score (CRISS) is described. **h** As in b except that GOI-NNG pair is based on g. **i** As in e, except that the 25 bins are based on GOI expression level (y-axis) and NNG expression level (x-axis). **j** Schematic indicating that readthrough Pol II from an upstream gene leads to read-in of downstream gene, causing the latter being upregulated. **k** CDF curves of CRISS vs. gene regulation for lowly expressed genes (left) and all genes (right). *P* values (K−S test) for comparisons of upregulated genes or downregulated genes with not regulated genes (gray lines) are indicated. Source data are provided as a Source Data file.

## APA analysis reveals mode of action of JTE-607 in PAS usage regulation

We found that the degree of 3′UTR lengthening after JTE-607 treatment, as measured by the difference in dPAS isoform to pPAS isoform abundance ratio, or Δlog$_2$(dPAS/pPAS), was a function of the distance between the two APA sites (Fig. 6a). For simplicity, the region between pPAS and dPAS was named alternative UTR, or aUTR, (Fig. 5a). Because pPAS is transcribed earlier than dPAS during transcription, a longer aUTR would favor pPAS usage. As such, the correlation between degree of 3′UTR lengthening and aUTR size suggests that APA sites

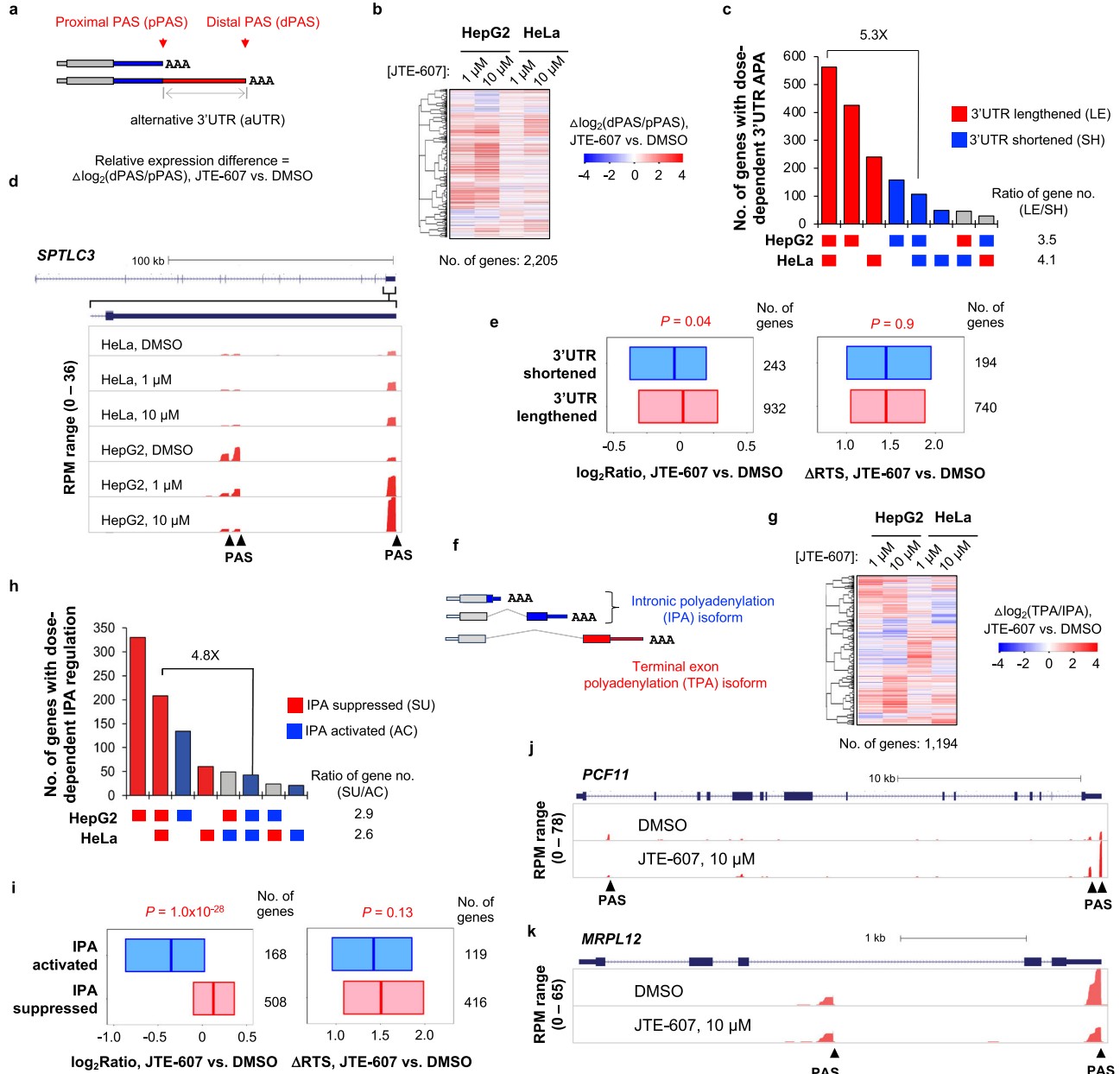

**Fig. 5 | Widespread APA isoform changes in HeLa and HepG2 cells treated with JTE-607. a** Schematic showing two 3′ UTR isoforms using proximal PAS (pPAS) and distal PAS (dPAS), respectively. The region between the two PASs is named alternative UTR (aUTR). **b** Heatmap showing relative expression difference of the top two 3′UTR isoforms (Δlog₂Ratio, dPAS isoform vs. pPAS isoform, JTE-607 vs. DMSO). **c** UpSet plot showing genes with dose-dependent 3′UTR isoform changes after JTE-607 treatment in HepG2 or HeLa cells. Data are based on b. Genes are divided into eight groups based on 3′UTR isoform regulation. Gray bars indicate different directions of regulation in the two cell lines. **d** An example gene *SPTLC3* showing dose-dependent 3′UTR lengthening in both HeLa and HepG2 cells. **e** Boxplots showing expression changes (left) and ΔRTS (right) for genes showing

lengthened 3′UTRs (red) or shortened 3′UTRs (blue) in HepG2 cells treated with 10 μM JTE-607. *P* values (Wilcoxon test) indicating significance of difference between the two gene groups are shown. Each box represents the 25th and 75th percentile values, with the median shown as a line. **f** Schematic of intronic polyadenylation (IPA) isoforms and 3′ terminal exon polyadenylation (TPA) isoforms. **g** As in b, except that data for TPA isoforms vs. IPA isoforms is shown. **h** As in **c**, except that IPA regulation data is shown. **i** As in **e**, except that IPA suppressed genes (red) and IPA activated genes (blue) are analyzed. **j** *PCF11* displays IPA isoform suppression and concomitant TPA isoform upregulation in HepG2 cells treated with 10 μM JTE-607. **k** As in **j**, except that *MRPL12* displays TPA suppression without IPA activation. Source data are provided as a Source Data file.

with higher usage levels tend to be more inhibited by JTE-607. Interestingly, we also found that this correlation was stronger in HepG2 cells than in HeLa cells (Fig. 6a), suggesting a cell type difference in CPA activity.

To further examine the CPA difference between HepG2 and HeLa cells, we profiled APA isoforms in these two cell lines. By comparing relative abundances of two APA isoforms, we found that HepG2 cells displayed a higher pPAS to dPAS isoform abundance ratio in general,

either when both APA sites were in the same 3′UTR (last exons only, 1.2-fold bias, Fig. 6b) or when the pPAS was in an intron and the dPAS was in the last exon (4.4-fold bias, Supplementary Fig. 5a), indicating that HepG2 cells have a general preference of pPAS usage as compared to HeLa cells. This result suggests that HepG2 cells have a higher CPA activity than HeLa cells. Interestingly, these two cell lines had a similar proliferation rate (Supplementary Fig. 5b), a feature previously associated with 3′UTR size difference between cells[16]. Regardless of the

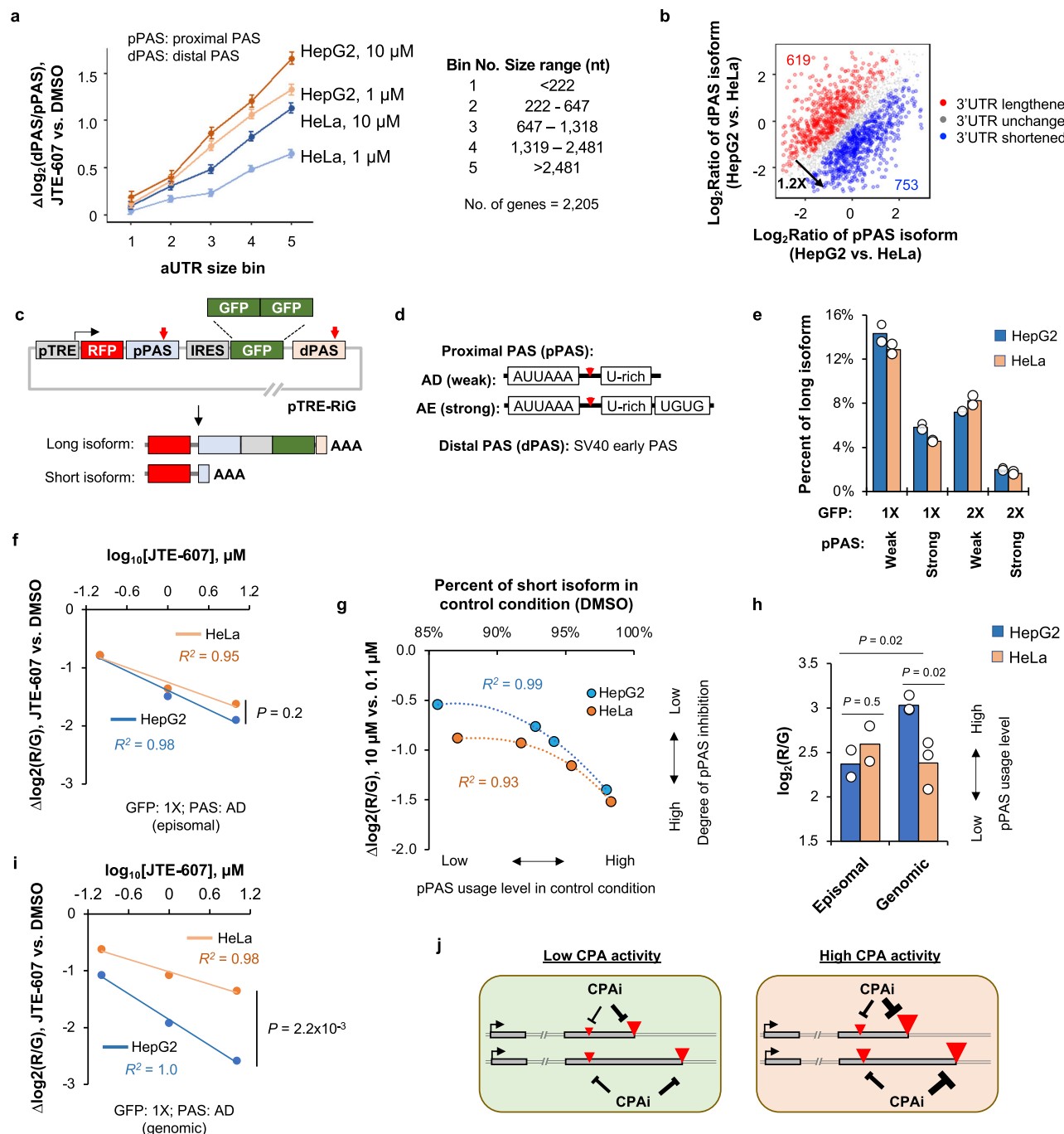

**Fig. 6 | APA reporter assays indicate that CPA inhibition by JTE-607 correlates with PAS usage level. a** The degree of 3′UTR lengthening elicited by JTE-607, as indicated by the relative expression difference of top two 3′UTR isoforms (Fig. 5a), is a function of aUTR size. All genes with detectable expression of 3′UTR isoforms ($n = 2205$) were divided into five equally sized bins based on their aUTR size. The aUTR size range for each bin is shown. Error bars are standard error of mean. **b** Scatter plot comparing 3′UTR isoform abundance difference between HepG2 and HeLa cells. Each dot represents a gene with two selected 3′UTR isoforms. Genes showing lengthened and shortened 3′UTRs in HepG2 compared to HeLa are in red and blue, respectively. **c** Schematic of the pTRE-RiG construct. TRE tetra-cycline response element, RFP red fluorescent protein, GFP green fluorescent protein, IRES internal ribosome entry site. As indicated, ratio of red to green fluorescent signals, or $\log_2(R/G)$, reflects the relative usage of pPAS vs. dPAS. **d** Schematic showing two versions of pPAS, i.e., AD and AE, with different PAS motifs. **e** Percentage of long isoform expressed in HepG2 or HeLa cells after

transfection of various constructs. **f** Reduction of $\log_2(R/G)$ value in HepG2 and HeLa cells transiently transfected with pTRE-RiG-AD (weak pPAS, 1× GFP) after JTE-607 treatments. Linear regression is based on averaged values of two biological replicates. $P$ value (t-test) comparing values for 10 μM JTE-607 data is indicated. **g** The degree of inhibition of pPAS usage by JTE-607, based on $\Delta\log_2(R/G)$ between 10 μM and 0.1 μM (y-axis), correlates with its usage level in the control condition (DMSO-treated), as indicated by the percent of short isoform expressed (x-axis). Polynomial regression is based on averaged values of two biological replicates. **h** Bar graph showing pPAS usage levels of pTRE-RiG-AD, indicated by $\log_2(R/G)$, in HeLa and HepG2 cells. Data for episomal form (transient transfection) or genomic form (integrated via piggyBac transposase) of the reporter are shown. $P$ values are based on $t$-test. **i** As in f, except that the construct was genome-integrated. **j** Schematic summarizing mode of action of JTE-607 (CPAi) in APA regulation. Source data are provided as a Source Data file.

underlying mechanism(s) responsible for the CPA difference between HeLa and HepG2 cells, the differential APA isoform regulation by JTE-607 in these two cell lines (Fig. 6a) suggests that PASs are more strongly inhibited by JTE-607 in cells with higher CPA activities.

To validate our hypotheses, we employed a reporter construct, named pTRE-RiG[44] (illustrated in Fig. 6c), in which pPAS usage leads to expression of a transcript encoding a red fluorescent protein (RFP), whereas dPAS usage leads to expression of a transcript encoding both RFP and green fluorescent protein (GFP). As such, the ratio of red fluorescent signal to green fluorescent signal, or $\log_2(R/G)$, reflects the relative usage of pPAS vs. dPAS (Fig. 6c). In addition, a promoter containing the tetracycline response element (pTRE), inducible by Doxycycline (Dox), drives the expression of reporter transcripts.

We used two pPASs with different strengths[17] and a constant, strong dPAS (SV40 early PAS, Fig. 6d). The relatively weaker pPAS (named AD) led to long isoform expression level of 13% (Fig. 6e), whereas a stronger one (named AE), containing additional UGUG motifs downstream of the PAS, gave rise to long isoform expression of 5% (Fig. 6d, e). In addition, increasing the distance between pPAS and dPAS by adding one additional copy of GFP (Fig. 6c) led to a 2-3-fold decrease of long isoform expression for both AD and AE constructs (Fig. 6e), confirming that the distance between APA sites positively impacts pPAS usage.

We next applied JTE-607 at various concentrations to HepG2 or HeLa cells transfected with different reporter constructs. As exemplified in Fig. 6f, a dose-dependent decrease of $\log_2(R/G)$ value could be observed in both HepG2 and HeLa cells, indicating that the reporter construct recapitulates pPAS inhibition by JTE-607 as observed with endogenous genes. Interestingly, we found that, in both HepG2 and HeLa cells, the pPASs with higher usage levels in control cells (DMSO, x-axis, Fig. 6g) were more strongly inhibited by JTE-607 ($\Delta\log_2[R/G]$ between 10 and 0.1 μM JTE-607 treatments, y-axis, Fig. 6g), with $R^2 = 0.99$ and 0.93 for HepG2 and HeLa cells, respectively (Fig. 6g), which is in good agreement with the regulatory trend observed with endogenous APA sites (Fig. 6a).

Because our reporter constructs showed only modest differences between HepG2 and HeLa cells in APA (Fig. 6e) or APA changes in response to JTE-607 (Fig. 6f), not in full accord with the data of endogenous genes, we wondered if inserting the plasmids into the genome would recapitulate endogenous gene regulation results better. To this end, we used a piggyBac transposase to insert one of the constructs, pTRE-RiG-AD, into the genomes of HepG2 and HeLa cells. Strikingly, the genome-integrated construct showed significantly higher pPAS usage in HepG2 cells than HeLa cells ($P < 0.05$, t-test, Fig. 6h). In addition, the genome-integrated constructs showed much stronger pPAS inhibition by JTE-607 in HepG2 cells than HeLa cells (note the slopes of fitted lines in Fig. 6i). Similar results were also obtained when we used a reporter construct designed for IPA analysis (Supplementary Fig. 5c). Again, a genome-integrated construct showed greater IPA suppression in HepG2 cells than HeLa cells (Supplementary Fig. 5d), a trend that was much subdued when the construct was used episomally by transient transfection (Supplementary Fig. 5e). Taken together, our reporter assays confirm that a higher PAS usage level renders more CPA inhibition by JTE-607 and, by the same logic, a PAS is inhibited to a greater degree when expressed in cells with higher CPA activities (illustrated in Fig. 6j).

## U937 cells and differentiated U937 cells with higher CPA activities respond to JTE-607 differently

To further examine how the CPA activity of a cell could impact JTE-607 functions, we used the human myeloid leukemia model cell line U937, which was previously found highly sensitive to JTE-607 for cell survival[36]. Its mouse xenograft was also used as an in vivo model for JTE-607's function in suppression of leukemia[39]. U937 cells are monocyte-like cells that can be differentiated into macrophage-like cells by Phorbol-12-Myristate-13-Acetate (PMA). For simplicity, the differentiated U937 cells are called U937MP cells (Fig. 7a). Compared to U937 cells, U937MP cells were significantly less proliferative (doubling time >200 h vs. 16 h for U937, Fig. 7b) and, consistently, U937MP cells are mostly in the G0/G1 phase (Supplementary Fig. 6a). In addition, U937MP cells were more tolerant of JTE-607 than U937 cells ($IC_{50} = 16.2$ μM vs. 0.4 μM, Fig. 7c).

Using the QuantSeq method for transcriptome analysis, we found that gene expression profiles are drastically different between U937 and U937MP cells (Supplementary Fig. 6b) and GO analysis of differentially expressed genes supported the view that U937MP cells are more differentiated and less proliferative (Supplementary Fig. 6c). Interestingly, we found that genes overall displayed shorter 3′UTRs in U937MP cells than in U937 cells (5.8-fold gene number difference between shortened and lengthened genes, Fig. 7d), suggesting the CPA activity increases during differentiation of U937 cells. Therefore, U937 and U937MP cells represent an isogenic system which could help unravel the relationships between CPA activity and JTE-607-elicited transcriptomic changes and cell death.

We treated U937MP and U937 cells with 1 μM JTE-607 and carried out chrRNA-seq and QuantSeq at 2 h and 7 h post treatments, respectively (Fig. 7e). Note that no cell death was discernable at 7 h post treatment for either cell line (Supplementary Fig. 7a). Overall, gene expression changes after JTE-607 treatment were correlated between these two cell types ($r = 0.53$, Pearson Correlation, Supplementary Fig. 7b) and were similar to those in HepG2 cells (Supplementary Fig. 7b). Interestingly, in contrast to their $IC_{50}$ difference, more genes were regulated in U937MP cells than in U937 cells (Fig. 7f). In addition, while more genes were downregulated than upregulated in both cell types, this trend was more obvious in U937MP cells (Fig. 7f). We also found that both cell types showed global 3′UTR lengthening after JTE-607 treatment (Fig. 7g), and the 3′UTR APA regulation was generally correlated between the two cell types ($r = 0.5$, Pearson Correlation, Supplementary Fig. 7c) and compared to HepG2 cells ($r > 0.5$, Pearson Correlation, Supplementary Fig. 7c). However, U937MP cells displayed more 3′UTR APA changes than U937 cells (Fig. 7g) and the degree of 3′UTR lengthening were more strongly correlated with aUTR size in U937MP cells than U937 cells ($P = 3.9 \times 10^{-5}$ for genes in bin 5, Wilcoxon test, Supplementary Fig. 7d).

Based on ΔRTS scores (JTE-607 vs. DMSO, chrRNA-seq data), we found that transcriptional readthrough after JTE-607 treatment increased much more in U937MP cells than U937 cells ($P = 1.1 \times 10^{-4}$, K–S test, Fig. 7h; Supplementary Fig. 7e). An example gene HNRNPD is shown in Fig. 7i. Using chrRNA and RT-qPCR with primer sets targeting regions downstream of the last PAS (readthrough region) and gene body (as a control), we found that while JTE-607 elicited transcriptional readthrough on HNRNPD in both U937 and U937MP cells (Fig. 7j), the latter was much stronger than the former (note the scale difference in Fig. 7j). Taken together, despite that U937 cells are more sensitive to JTE-607 than U937MP cells in cell survival, the latter displays much stronger response in transcriptomic changes than the former, including gene expression, transcriptional readthrough, and APA. This result is in good agreement with the notion that cells with higher CPA activities (U937MP) tend to have more PAS inhibition by JTE-607. On the other hand, this result indicates that the JTE-607-elicited transcriptomic changes are not the sole factor leading to cell death.

## FIP1 overexpression leads to global activation of CPA

To address the conundrum of cell death vs. transcriptomic change by JTE-607, we set out to establish an isogenic system in which the CPA activity is tunable while there are no other major changes to the cell. Studies from our lab and others have shown that knockdown of the core CPA factor FIP1 leads to global 3′UTR lengthening, indicating its importance for controlling the CPA activity[45,46]. FIP1 has at least

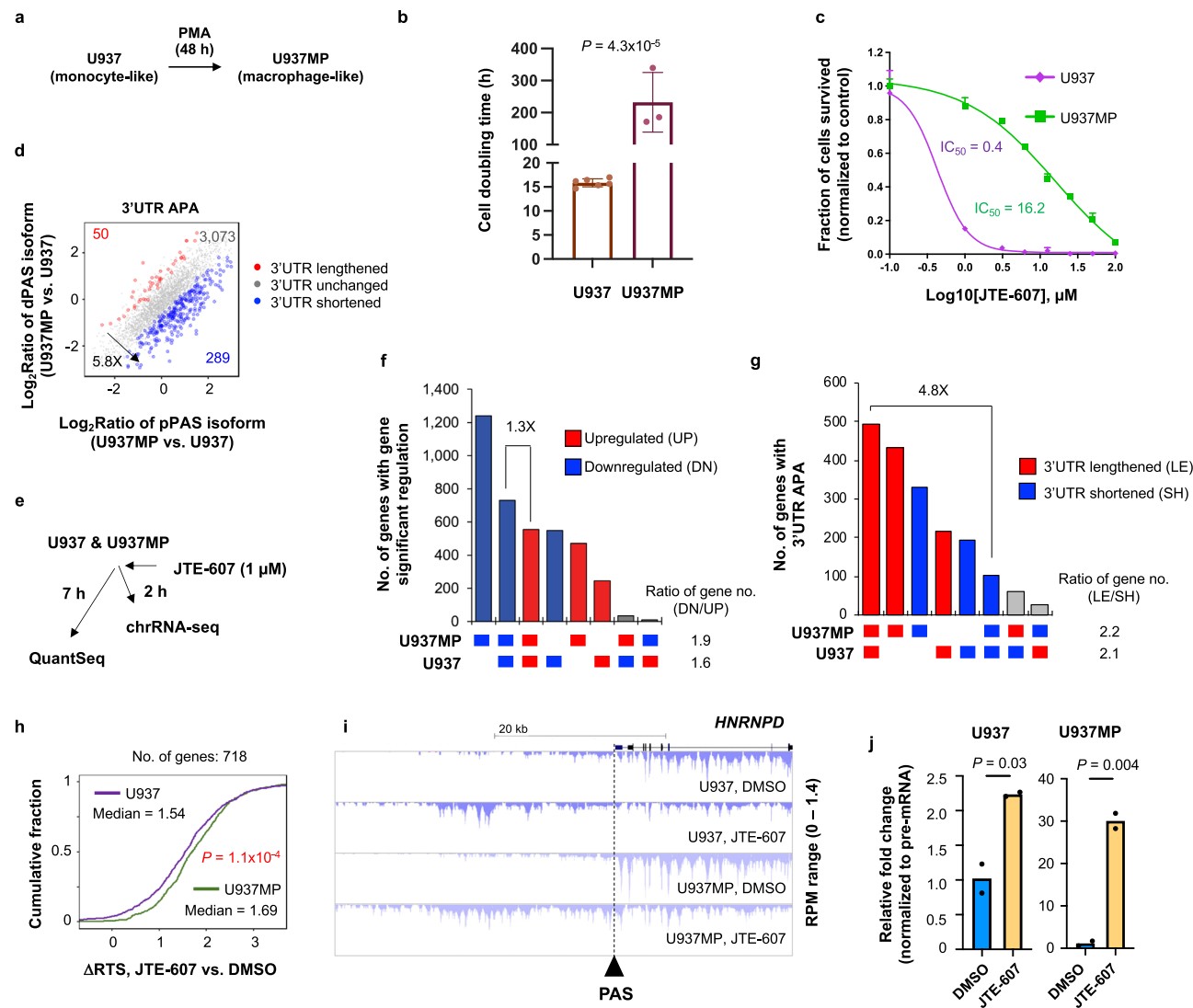

**Fig. 7 | U937 cells respond to JTE-607 differently after differentiation.**
**a** Schematic showing differentiation of monocyte-like U937 cells to macrophage-like U937MP cells after treatment of 10 ng/mL Phorbol-12-Myristate-13-Acetate (PMA) for 48 h. **b** Cell doubling time analysis of U937 and U937MP cells. Error bars are standard deviation based on biological replicates ($n = 6$ and $= 3$ for U937 and U937MP cells, respectively). *P* value (*t*-test) for significance of difference in cell doubling time is indicated. **c** $IC_{50}$ analysis of U937 and U937MP cells in response to JTE-607, as determined by the AlamarBlue assay. Error bars are standard deviation ($n = 3$). **d** Scatter plot comparing 3′UTR isoform abundance difference between U937MP and U937 cells. Each dot is a gene with two 3′UTR isoforms. Genes with 3′ UTR significantly lengthened and shortened in U937MP cells as compared to U937 cells are shown in red and blue, respectively (see Methods for gene selection details). The number of genes for each group is indicated, and so is their ratio. **e** Schematic showing QuantSeq and chrRNA-seq analyses of U937 and U937MP cells treated with JTE-607. **f** UpSet plot showing numbers of genes with significant expression changes in U937 cells and/or U937MP cells after JTE-607 treatment (BH-adjusted $P < 0.05$, Fisher's exact test; fold change ≥2). Genes are divided into eight groups based on regulations in the two cell lines. Gray bars indicate different directions of regulation between the two cell lines. The ratio of number of down-regulated genes to that of upregulated genes is indicated. **g** As in **f**, except that 3′ UTR APA data are shown. **h** CDF curves of ΔRTS in U937 and U937MP cells. The number of genes analyzed, the median value of each cell line, and *P* value (K–S test) for significance of difference between the two cell lines are indicated. **i** An example gene *HNRNPD* showing readthrough difference between U937 and U937MP cells. **j** qRT-PCR analysis showing JTE-607-elicited transcriptional readthrough of gene *HNRNPD* in U937 (left) or U937MP (right) cells ($n = 2$). chrRNA signal in read-through region was normalized to that in gene body. *P* values (*t*-test) for significance of difference are indicated. Source data are provided as a Source Data file.

20 splicing isoforms documented in the NCBI database (Supplementary Fig. 8a). We cloned the open reading frame of the longest isoform (NM_030917.4), encoding 595 amino acids, into a lentiviral vector under the control of a TRE-containing promoter (Fig. 8a). On the same vector, an internal ribosome entry site (IRES)-enhanced GFP (EGFP) or IRES-blue fluorescent protein (BFP) sequence was placed after the FIP1 sequence so that the expression level of this exogenous FIP1 could be tracked by EGFP or BFP levels (Fig. 8a, d).

Using HEK293T cells transduced with the inducible FIP1 vector (named HEK293T/iFIP1 cells), we found that induction of FIP1 overexpression (OE) for 24 h led to global 3′UTR shortening

(2.5-fold bias for pPAS isoform vs. dPAS isoform, Fig. 8b) as well as IPA activation (4.4-fold bias for IPA isoform vs. TPA isoform, Fig. 8c), indicating that FIP1 OE elevates CPA activity in the cell, leading to general activation of pPAS usage. In addition, using a genome-integrated reporter construct pTRE-RiG-AD (Fig. 6c), we found that cells with high blue fluorescent signals had a high log2(R/G) value as compared to cells with low or no blue fluorescent signals (Fig. 8e), indicating that FIP1 expression leads to more usage of pPAS of the reporter gene. Importantly, cells with high FIP1 OE tended to have greater dose-dependent inhibition of pPAS usage as compared with cells with low or no FIP1 OE (see slope differences in Fig. 8f),

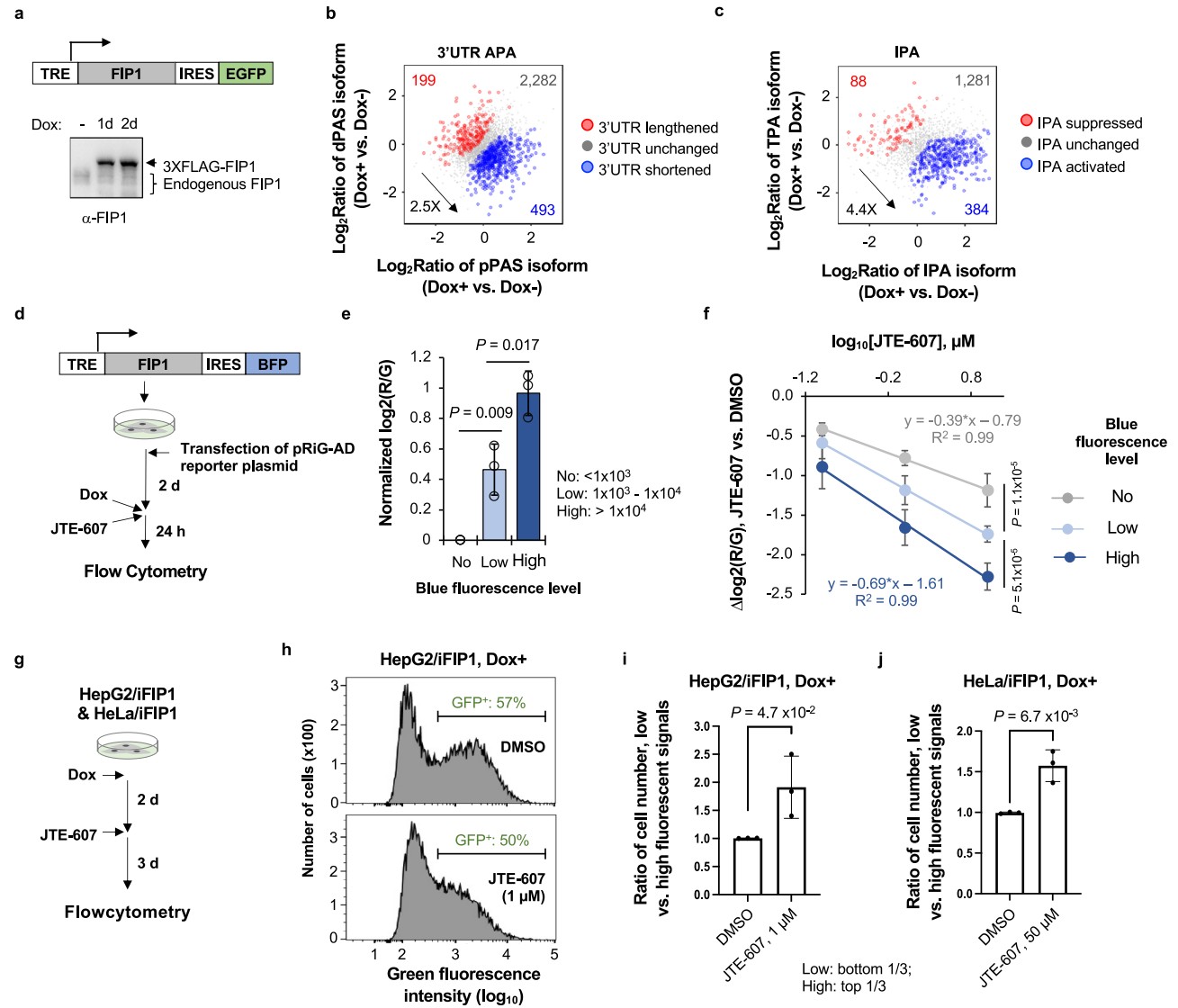

**Fig. 8 | FIP1 overexpression leads to global activation of CPA activity.**
**a** Schematic of a lentiviral vector containing an inducible FIP1 (top) and Western blot (bottom) showing induction of FIP1 in HEK293T cells by 2 µg/mL Dox for one or two days. **b** Scatter plot comparing 3′UTR isoform abundance difference between HEK293T cells with FIP1 induction (Dox+, 24 h) and those without (Dox−). Each dot represents a gene with two selected 3′UTR isoforms (as shown in Fig. 5a). **c** As in b, except that IPA isoforms are analyzed. See Fig. 5f for IPA and TPA isoform definitions. **d** Schematic of reporter assay to examine APA regulation by FIP1 overexpression (OE) using pTRE-FIP1-IRES-BFP. **e** Bar graph indicating the level of pPAS usage in HEK293T cells with the blue fluorescent signal tracking FIP1 expression level. Cells showing no, low, or high blue signals were analyzed for pPAS usage, as indicated by log2($R/G$). Error bars are standard deviation based on three biological replicates. $P$ values ($t$-test) for significance of difference between comparing groups are indicated. **f** Inhibition of pPAS usage by JTE-607 in cells with

different levels of FIP1 OE. Linear regression formula and $R^2$ values are shown for the cell groups with no or high BFP signals. Error bars are standard deviation of three biological replicates. $P$ value ($t$-test) for significance of difference in $\Delta$log2($R/G$) (10 µM JTE-607 vs. DMSO) between cell groups are indicated. **g** Schematic of the cell competition assay, in which HepG2 or HeLa cells with inducible FIP1 OE (named HepG2/iFIP1 and HeLa/iFIP1, respectively) are subject to JTE-607 treatment, followed by flow cytometry analysis to examine green or blue fluorescent signals of surviving cells. **h** Histograms showing distribution of fluorescence signals of HepG2 cells after DMSO (top) or JTE-607 (bottom) treatment. **i** Bar graph showing ratio of number of cells with low (bottom 1/3) to high fluorescent signals (top 1/3) in HepG2 cells treated with 1 µM JTE-607. Error bars are standard deviation of three biological replicates. $P$ ($t$-test) for significance of difference is indicated. **j** As in **i**, except that HeLa/iFIP1 cell data are shown with 50 µM JTE-607. Source data are provided as a Source Data file.

supporting the notion that the degree of PAS inhibition by JTE-607 correlates with its usage level.

We next transduced HepG2 and HeLa cells with the FIP1 OE vector and subjected the cells (named HepG2/iFIP1 and HeLa/iFIP1, respectively) to JTE-607 treatments. After three days of treatment, we analyzed cells for their fluorescent signals (Fig. 8g). We reasoned that if FIP1 OE made cells more sensitive to JTE-607, cells with low fluorescent signals, corresponding to low FIP1 expression levels, would be more abundant than cells with high fluorescent signals. Indeed, we found that HepG2 cells with FIP1 OE had a higher ratio of low fluorescent cells

to high fluorescent cells after JTE-607 (1 µM) treatment as compared to DMSO-treated cells (Fig. 8h, i). A similar result was obtained with HeLa cells with FIP1 OE after JTE-607 (50 µM) treatment (Fig. 8j and Supplementary Fig. 8b). Together, these data indicate that FIP1 overexpression, which elevates CPA activity in the cell, makes cells more sensitive to JTE-607.

### FIP1 overexpression sensitizes cells for CPAi
To further examine the impact of FIP1 OE on cell survival, we generated two U937 cell clones containing inducible exogenous FIP1,

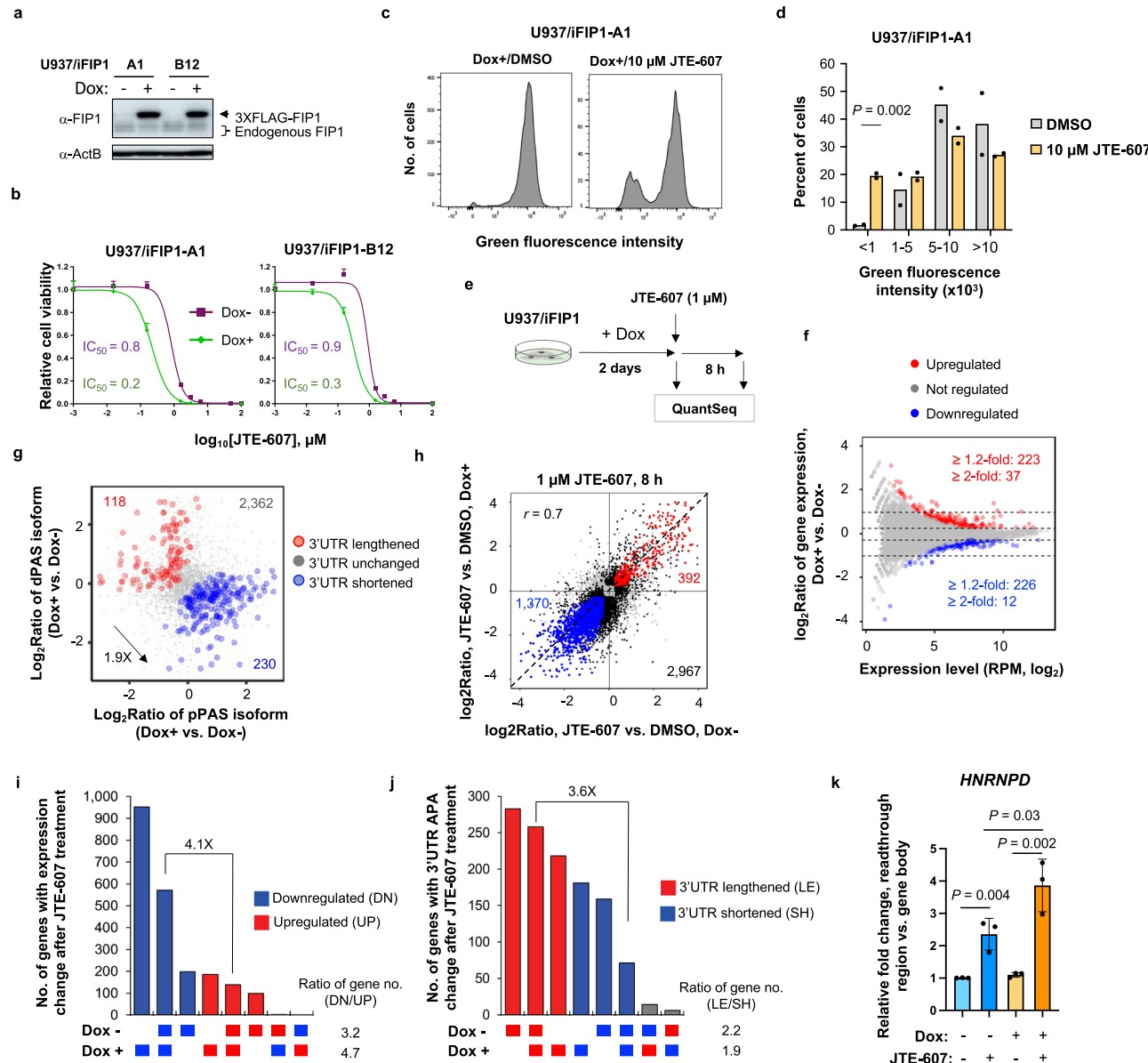

**Fig. 9 | FIP1 overexpression leads to greater cell sensitivity to JTE-607.**
**a** Western blot showing induction of FIP1 in two U937/iFIP1 clones (A1 and B12).
**b** IC$_{50}$ analysis of U937/iFIP1-A1 and -B12 cells treated with JTE-607. Error bars are standard deviation ($n = 3$). **c** Histograms showing distribution of fluorescence signals of U937/iFIP1-A1 after DMSO (left) or JTE-607 (right) treatment. **d** Bar graph showing percent of U937/iFIP1-A1 cells with different green fluorescence levels after the cell competition assay with 10 μM JTE-607 ($n = 2$). $P$ is based on $t$-test. **e** Schematic showing transcriptome analysis of U937/iFIP1 cells with or without FIP1 induction (Dox+ or Dox−, respectively) followed by JTE-607 treatment. **f** Gene expression changes after FIP1 induction for 48 h. The number of genes with significant expression changes (BH-adjusted $P < 0.05$, Fisher's exact test; fold change ≥1.2 or ≥2) are indicated. A total of 13,056 genes were analyzed. **g** 3′UTR shortening in U937/iFIP1 cells after FIP1 induction. See Fig. 8b for description of the plot. **h** Comparison of JTE-607-elicited gene expression changes in U937/iFIP1 cells in Dox+ vs. Dox− conditions. Genes with significant expression changes in both conditions (BH-adjusted $P < 0.05$, Fisher's exact test; fold change ≥1.2) are colored

with red (both upregulated) or blue (both downregulated). Genes in black are significant in only one condition and genes in gray are not significant in either condition. A total of 11,311 genes were analyzed. Pearson correlation coefficient ($r$) is indicated. **i** UpSet plot showing numbers of genes with significant JTE-607-elicited expression changes (BH-adjusted $P < 0.05$, Fisher's exact test; fold change ≥2) in U937/iFIP1 cells in Dox+ and/or Dox− conditions. Gray bars indicate different directions of regulation between the two conditions. The ratio of number of downregulated genes to that of upregulated genes is indicated. **j** As in **i**, except that JTE-607-elicited 3′UTR APA events are compared between U937/iFIP1 cells in Dox+ vs. Dox− conditions. **k** qRT-PCR analysis showing JTE-607-elicited transcriptional readthrough of gene *HNRNPD* in U937/iFIP1 cells with or without FIP1 induction. chrRNA signal in readthrough region was normalized to that in gene body. Error bars are standard deviation ($n = 3$). $P$ values ($t$-test, one-tailed) for significance of difference in readthrough are indicated. Source data are provided as a Source Data file.

named U937/iFIP1-A1 or -B12 cells (Fig. 9a). These two clones had a 3- or 4-fold decrease of IC$_{50}$ value for JTE-607 when FIP1 was induced (Dox+) as compared to cells without FIP1 induction (Dox−, Fig. 9b), indicating that, similar to HeLa/iFIP1 and HepG2/iFIP1 cells, increased CPA activity through FIP1 OE makes U937 cells more sensitive to JTE-607. In addition, surviving U937 cells after a high

dosage of JTE-607 treatment (10 μM, which killed most cells) showed a much higher percentage of cells with low fluorescent signals than DMSO-treated control cells (Fig. 9c, d and Supplementary Fig. 9a–c).

We next examined the transcriptomic changes in U937/iFIP1 cells with vs. without Dox induction (Fig. 9e). After 48 h of Dox induction,

only a small number of genes displayed expression changes (449 and 49 with fold change ≥1.2 and ≥2, respectively, Fig. 9f), indicating that FIP1 OE has only a limited effect on mRNA expression levels. However, consistent with the HEK293T cell data, more genes showed shortened 3′UTRs than lengthened 3′UTRs (Fig. 9g) and IPA activation (Supplementary Fig. 9d), indicating increased CPA activity after FIP1 OE. As expected, JTE-607 elicited widespread gene expression changes in U937/iFIP1 cells (Fig. 9h). While the gene expression changes are largely correlated ($r = 0.7$, Pearson Correlation, Fig. 9h) and more genes were downregulated than upregulated in both Dox− and Dox+ conditions ($P < 0.05$, Fisher's exact test, fold change ≥2, Fig. 9i), U937 cells with FIP1 OE displayed much more regulated genes, especially in the downregulation group (Fig. 9i). In addition, commonly regulated genes between Dox+ and Dox− conditions showed general trends of 3′UTR lengthening (Fig. 9j) and IPA suppression (Supplementary Fig. 9e). Based on the chrRNA analysis of *HNRNPD* gene, cells with FIP1 OE had greater transcriptional readthrough after JTE-607 treatment than cells without OE (Fig. 9k). Taken together, our data based on U937 cells with FIP1 OE indicate that elevation of CPA activity leads to higher sensitivity to JTE-607 in cell survival and greater response in transcriptomic disturbance.

## CPAi leads to S phase crisis and is synergistic with DNA repair inhibitors

JTE-607 was previously reported to cause DNA damage in Ewing's sarcoma cells[36] and cell cycle arrest in the S phase and apoptosis in U937 cells[47]. With the U937/iFIP1 cells, we next asked whether CPA activity elevation through FIP1 OE could cause more DNA damage by JTE-607. To this end, we first examined the γH2A.X level, an indicator of DNA damage, in U937/iFIP1 cells +/− Dox induction. We found that JTE-607 treatment (1 μM, 24 h) increased the percent of γH2A.X positive cells by 4.5-fold in Dox− condition but 11.7-fold in Dox+ condition, significantly higher than the former ($P = 0$, Chi-squared test, Fig. 10a, b). Note that Dox induction alone without JTE-607 treatment had no effect on γH2A.X signals ($P = 0.2$, Chi-squared test, Fig. 10a, b).

We next examined cell cycle phases by propidium iodide (PI) staining 24 h post JTE-607 treatment (0.4 μM, Fig. 10c and Supplementary 10a). We found that while Dox− and Dox+ cells did not differ in their cell cycle phase profiles (Fig. 10c), JTE-607 treatment markedly enriched cells in the S phase (representative data for the two clones shown in Fig. 10c and Supplementary Fig. 10a). The percent of cells in the S phase in JTE-607-treated cells increased by 1.6-fold and 1.4-fold in Dox− and Dox+ conditions, respectively, as compared to their respective untreated cells (Fig. 10d and Supplementary 10b), indicating that JTE-607 elicits S phase crisis in these cells. This trend was also discernable, albeit much less obvious, at 12 h post JTE-607 treatment (Supplementary Figs. 10d and 10e). We also noticed that the fraction of cells with fragmented DNA were much higher in Dox+ condition than Dox− condition after JTE-607 treatment (the <2 N group with low PI signals, Fig. 10c, d and Supplementary Fig. 10a), which may contribute to a slightly smaller fraction of S phase cells in Dox+ & JTE-607-treated condition as compared to Dox− & JTE-607-treated condition. The presence of cells with fragmented DNA may also suggest S phase crisis-elicited apoptosis. Also of note is that the ratio of cells in the 1st half of S phase to those in the 2nd half in Dox+ condition was much higher than that in Dox− condition after JTE-607 treatment ($P = 0.01$, t-test, Fig. 10e and Supplementary 10c), indicating more severe S phase crisis in FIP1 OE cells. It is noteworthy that cells in Dox+ and Dox− conditions did not differ significantly in their proliferation rate (Supplementary Fig. 10f), indicating that FIP1 OE per se has no influence on the cell cycle. Together, these results indicate that a higher CPA activity of a cell leads to greater DNA damage and S phase crisis after JTE-607 treatment.

The DNA damage and S phase crisis data further prompted us to examine whether JTE-607 would be synergistic with anti-cancer compounds that function through DNA damage repair pathways. To this end, we treated U937/iFIP1 cells with JTE-607 together with hydroxycamptothecine (topoisomerase I inhibitor, or TOPIi), mitoxantone (topoisomerase II inhibitor, or TOPIIi), or BAY 1895344 (ataxia telangiectasia and Rad3 related [ATR] kinase inhibitor, or ATRi). Interestingly, based on the Bliss independence model, we observed pronounced synergic effects on cell death (synergy score >10) between JTE-607 and all these compounds in U937/iFIP1 cells, except for hydroxycamptothecine in Dox− cells (Fig. 10f, g, and Supplementary Fig. 11a). Importantly, the synergy scores were markedly higher in Dox+ conditions than in Dox− conditions (note the 95% confidence intervals indicated in Fig. 10f), indicating that the synergistic effects are more potent in cells with higher CPA activities.

We next repeated the synergy analysis in U937 cells with an additional set of TOPIi (Topotencan), TOPIIi (Teniposide), and ATRi (AZD6738) compounds (Fig. 10h, and Supplementary Fig. 11b). Again, we found that all TOPIi, TOPIIi, and ATRi compounds had synergy scores > 10 (Bliss method) with JTE-607, further confirming that CPAi is synergistic with inhibitors of DNA damage repair pathways. In line with these data, we also found that DNA synthesis inhibitors, cytarabine hydrochloride and floxuridine, were synergistic with JTE-607 in causing U937 cell death (supplementary Fig. 12). Together, our data indicate that JTE-607 causes S phase crisis and DNA damage in proliferative cells. This effect might be due to the collision between readthrough Pol II and replication fork, i.e., transcription-replication conflicts[48]. As such, JTE-607 is synergistic with chemotherapy drugs that inhibit DNA repair pathways and DNA synthesis, raising the possibility of using CPAi as an adjunct treatment in suppressing cancer cells with high proliferation rates.

## Discussion

In this study, we examine in different cell contexts the mode of action of JTE-607, the first known CPAi compound in mammalian cells. JTE-607 treatment leads to transcriptomic disturbance, attributable to transcriptional readthrough and APA site usage changes (Fig. 10i). Importantly, these CPAi effects are more potent in cells with high CPA activities, consistent with the notion that high PAS usage levels render more inhibition by JTE-607. In addition, JTE-607 treatment leads to DNA damage and S phase crisis, due likely to conflicts between readthrough Pol II and DNA replication. This is in line with a previous report showing that a proper CPA activity prevents replication stress-associated genome instability[49]. Consistently, JTE-607 is synergistic with DNA damage repair-based chemotherapeutic drugs, such as TOPIi, TOPIIi, and ATRi. Therefore, CPA activity and proliferation rate are two cellular determinants of JTE-607-mediated cell death (Fig. 10i), and CPAi could be a promising adjunct treatment to other chemotherapy drugs in suppressing certain cancers.

Our comparison of U937 and U937MP cells is particularly revealing about the relative importance of CPA activity vs. cell proliferation for JTE-607-mediated cell death. The quiescent U937MP cells have a higher CPA activity than proliferative U937 cells, which is in line with a recent report showing increased CPA factor expression in macrophage differentiation[50]. As such, U937MP cells display greater transcriptomic distances by JTE-607 than U937 cells. However, U937MP cells have a higher $IC_{50}$ than U937 cells, indicating that proliferation rate is a more potent determinant than CPA activity for JTE-607-mediated cell death. On the other hand, because the $IC_{50}$ of U937MP cells is still relatively low, compared to HeLa cells for example, we conclude that transcriptomic distance can also be lethal to the cell. We cannot, however, pinpoint which genes whose disturbances are the most critical for cell survival among the widespread transcriptomic changes.

JTE-607-meidated CPAi resembles that of *PCF11* KD, underscoring commonalities among CPA factor inhibitions. Interestingly, IPA of *PCF11* is strongly inhibited by JTE-607, in line with the notion that the IPA site of *PCF11* is a biological sensor of cellular CPA activity[27,43]. On

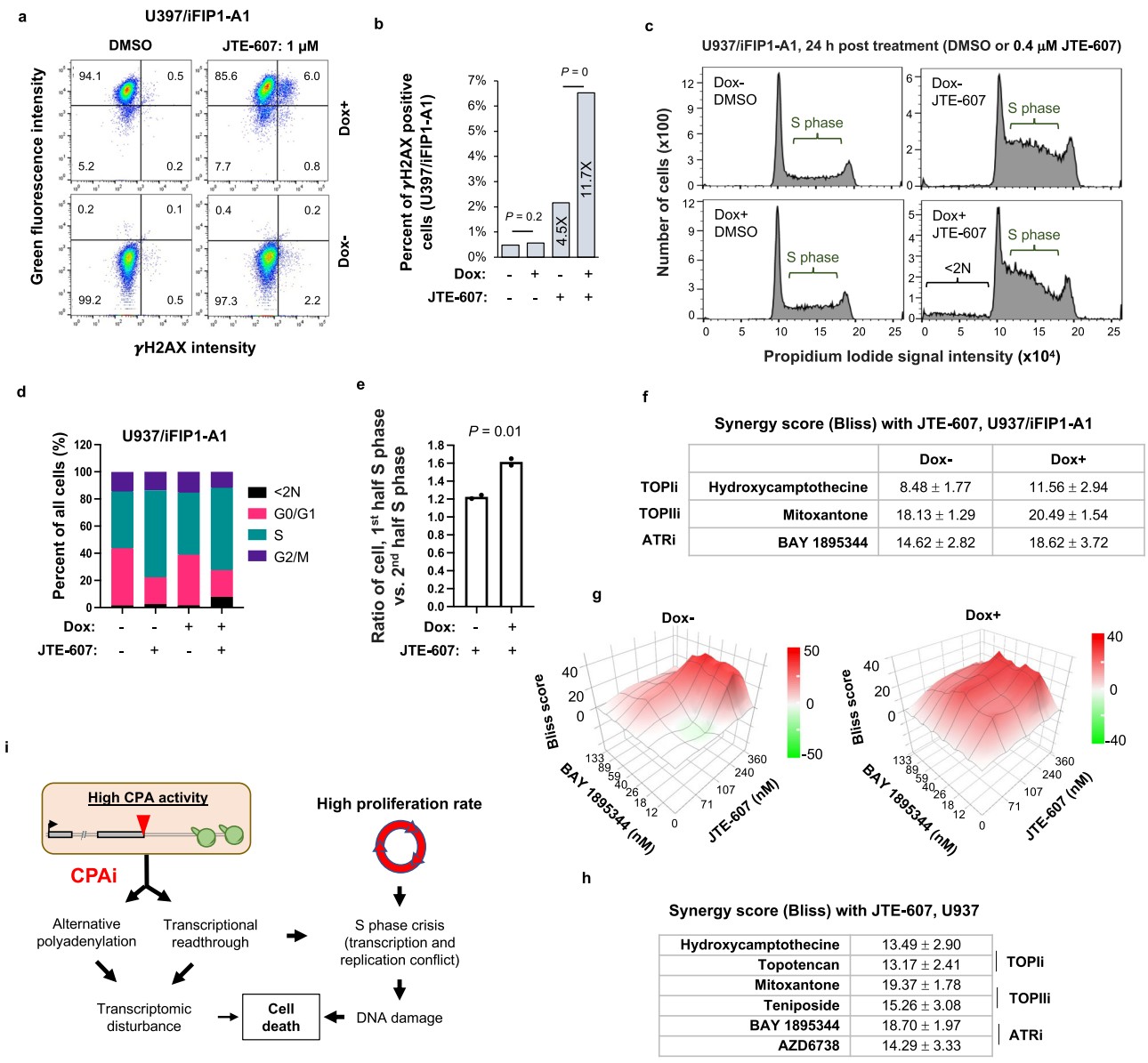

**Fig. 10 | JTE-607 leads to DNA damage and S phase crisis. a** Representative flow cytometry data of γH2A.X expression levels in U937/iFIP1 cells in Dox− or Dox+ conditions after JTE-607 treatment for 24 h. **b** Fold change of γH2A.X signals in U937/iFIP1-A1 cells with or without FIP1 induction after JTE-607 or DMSO treatment. *P* value is based on the Chi-squared test. Fold change is based on JTE-607-treated vs. DMSO-treated samples in Dox− and Dox+ conditions, respectively. **c** Cell cycle analysis using U937/iFIP1-A1 cells. Data are based on 24 h post JTE-607 treatment. **d** Percent of cells in different cell cycle phases as based on data in **c**. **e** Ratio of cells in the 1st half S phase to the 2nd half S phase after JTE-607 treatment in U937/iFIP1 cells in Dox+ vs. Dox− conditions (*n* = 2). *P* value (*t*-test) for significance of difference is indicated. **f** Synergy scores for JTE-607 with another indicated compound in U937/iFIP1 cells in Dox+ or Dox− conditions. Synergy scores were calculated by the SynergyFinder program using the Bliss method. The

'+/− value' indicates 95% confidence interval (two replicates). The function of each compound is indicated, namely, TOPIi, TOPIIi, and ATRi. **g** Representative synergy score plots generated by the SynergyFinder program for JTE-607 in combination with BAY 1895344 in U937/iFIP1-A1 cells in Dox− (left) or Dox+ (right) conditions. **h** As in **f**, except that synergy analysis was carried out in U937 cells (two replicates). **i** A model summarizing mode of action of JTE-607 (CPAi). Alternative polyadenylation and transcriptional readthrough lead to transcriptomic disturbance, the extent of which correlates with the CPA activity of a cell. Transcriptional readthrough also leads to S phase crisis, due likely to transcription-replication conflict, and then DNA damage, which is exacerbated by a high cell proliferation rate. CPAi causes cell death through both transcriptomic disturbance and DNA damage. Source data are provided as a Source Data file.

the other hand, the similarity between JTE-607 and *PCF11* KD raises the possibility that PCF11 might also be a good target for suppression of certain cancers[34,36,39]. In this vein, it is worth noting that PCF11 inhibition was shown to suppress neuroblastoma[51] and mutations in the noncoding region of *PCF11* gene have been implicated as pan-cancer drivers[52]. The question remains, however, as to whether inhibition of other core CPA factors would lead to similar effects. It is notable that we previously carried out systematic siRNA-based KD of core CPA factors and did not observe strong APA regulation in C2C12 myoblast

cells with CPSF-73 KD[45], suggesting that compensatory mechanisms could be in play to mitigate KD effects over time. It is therefore important in future studies to examine CPA factor inhibition in a short time window to analyze effects.

The top two features associated with JTE-607-mediated gene regulation are GC content and RNA stability score, both of which negatively correlated with gene expression changes. While these two features themselves are correlated, they have independent contributions to gene expression regulation. This notion is

supported by the fact that removal of both features causes a substantial decrease of $R^2$ value than does one feature alone. High GC content was previously found to negatively correlate with transcriptional elongation rate[53,54]. Because slower transcriptional elongation may enhance PAS usage[55,56], much like increasing the distance between APA sites, it is possible that the CPA activity is higher in high GC content regions, leading to greater CPAi on genes in these regions. This effect may be further enhanced by high gene densities, which tend to correlate with high GC contents. On the other hand, it is conceivable that genes in high density regions may have evolved to have high CPA activities to suppress Pol II readthrough-elicited transcriptomic disturbances, ensuring proper gene expression in normal conditions.

It is intriguing that genes having higher RNA stability scores are more likely to be downregulated after JTE-607 treatment. On the one hand, this result underscores the difference between CPAi and general transcriptional inhibition, such as by Actinomycin D, because the latter would suppress expression of short-lived RNAs to a greater degree. On the other, it may indicate that the transcripts with higher stability may be associated with greater efficiency in pre-mRNA 3′ end processing. How these two aspects of mRNA life are interconnected needs to be further examined in the future.

Our work is based on analysis of genes that express poly(A)+ RNA. CPSF-73 is also responsible for 3′ end processing of replication-dependent histone pre-mRNAs, whose mature forms do not have the poly(A) tail[57]. In fact, a recent study of pancreatic cancers showed drastic downregulation of these histone proteins after JTE-607 treatment[41]. Therefore, JTE-607-mediated inhibition of histone transcripts could further exacerbate S phase crisis, adding another layer of its function in suppressing proliferative cells. It should be noted, however, that the difference between FIP1 induced and noninduced cells in this study should not be affected by regulation of these histone genes, because FIP1 is not involved in 3′ end processing of replication-dependent histone genes[57].

Transcriptional readthrough leads to production of DoG transcripts[28,30], which has been shown in several biological conditions, such as cellular stress[29], viral infection[32], and certain cancers[33]. In addition, a growing number of molecular mechanisms have been implicated in control of transcriptional readthrough, such as Histone H3 lysine 36 methylation[33], BRD4-mediated Pol II elongation[58], the termination factor XRN2[59] and the Integrator complex[60]. Our data show that, when transcript abundance is controlled, similar levels of transcriptional readthrough are observed for Pan-DoGs vs. other genes, indicating that JTE-607, or perhaps CPAi in general, causes general readthrough across genes. On this note, it is noteworthy that JTE-607 may not be a pan-PAS inhibitor, as indicated by a recent study by Yongsheng Shi's group[61]. As such, some PASs are refractory to JTE-607. Further optimization of the compound is needed to improve its universality in PAS inhibition.

## Methods

### Plasmids

pCW2-hPGK-BSD-rtTA, used for generation of Tet-On cells, was constructed by using pCW-Cas9-BSD as a template. The Cas9 sequence was removed by using Cla I and BamH I, followed by blunt end generation with DNA polymerase I (Klenow fragment) and ligation with T4 DNA ligase. pCMV-RiG, pCMV-RiG-AD and pCMV-RiG-AE were constructed previously[22,44]. An mNeonGreen sequence from pHAGE-IRES-puro-NLS-dPspCas13b-2xmNeongreen-NLS-3xFlag (Addgene #: 132402) was amplified by using PCR and was cloned into the pCMV-RiG (XhoI and BsrGI) vector to make pCMV-RiG (1xmNeonGreen) by using the NEBuilder kit (NEB). A second mNeonGreen sequence was inserted to pCMV-RiG (1xmNeonGreen) to make pCMV-RiG (2xmNeonGreen) with NEBuilder. To construct PB-TRE-RiG-AD, the RiG-AD fragment from pCMV-RiG-AD was

amplified and subcloned into a vector derived from PB-CAG-BGHpA (Addgene #92161). pTRE-RiniG-1600-TT was previously constructed[62] and was used to generate PB-TRE-RiniG-1600-TT. The FIP1 ORF was cloned into pCW vector by PCR using the plasmid MIP-HA-FIP1 (a gift from Dr. Dong-Er Zhang, UCSD) as a template. PCR primer sequences are shown in Supplementary Table 1.

### Stable cell line construction

Tet-On cells were generated by transduction of pCW2-hPGK-BSD-rtTA into the cell. PB-TRE-RiG-AD and PB-TRE-RiniG-1600-TT were integrated into the genomes of HeLa and HepG2 Tet-On cells by using the piggyBac transposase. The transposase-containing plasmid Hypo-MDM-PB7-CMV was used for co-transfection. Cell selection started one day post transfection and was carried out in Hygromycin B (400 μg/mL) for at least 4 days. Cells without transposase were used as a control for cell selection. Lentiviral vectors were used to construct stable cell lines with inducible expression of FIP1. Briefly, HEK293T cells grown on a 6-cm dish were transfected using lipofectamine 3000 (Thermo Fisher) with the plasmids psPAX2, pMD2.G and pTRE-3xFlag-FIP1-IRES-EGFP-BSD at the ratio of 3:1:2 and the total amount of 6.88 μg DNA. Cell culture media collected at 24 h and 48 h post transfection were combined for concentration by ultracentrifugation. Target cells seeded in a 12-well plate were incubated with concentrated virus at 250 μL per well, together with 750 μL complete media containing polybrene (final conc. of 8 μg/mL). Cell selection was carried out with 10 μg/mL Blasticidin for at least 4 days. Selected cells were tested for reproducible, induction of FIP1 expression in each experiment.

### Cell culture and treatments

HEK293T, HeLa and HepG2 cells were cultured in Dulbecco's modified Eagle's medium (DMEM) supplemented with 10% fetal bovine serum (FBS) and 1% penicillin/streptomycin at 37 °C and were supplied with 5% $CO_2$. U937s were cultured in RPMI-1640 with 10% FBS and 1% penicillin/streptomycin. Macrophage-like U937 (U937MP) cells were differentiated from U937 cells by using 10 ng/mL PMA in RPMI-1640 complete media for 48 h, followed by culturing in fresh RPMI-1640 complete media and resting overnight before use. For QuantSeq analysis, HeLa and HepG2 cells were treated with DMSO or JTE-607 (1 μM or 10 μM) for 8 h before harvest for total RNA extraction. U937 and U937MP cells were treated with DMSO or 1 μM JTE-607 for 7 h before total RNA extraction. For chromatin RNA (chrRNA) sequencing, HeLa, HepG2, U937 and U937MP cells were treated with DMSO or 1 μM JTE-607 for 2 h before cells were collected for chromatin RNA extraction. Induction of FIP1 expression was carried out by using 2 μg/mL Dox.

### Cell doubling time and viability analysis

Cells were seeded in a 12-well plate at low density (10–20% confluency). Cells were harvested and counted by using the Countess II automated cell counter (Thermo Fisher Scientific), with the trypan blue dye to indicate cell viability. Cell doubling time was calculated with the GraphPad program, using multiple biological replicates.

### Cell survival assay

Cells seeded into a 96-well plate were treated with JTE-607 at various conc. Three replicates were used for each cell type. Control wells were treated with DMSO. After three days of treatment, AlamarBlue reagent (Thermo Fisher Scientific, Cat #: DAL1100) was added into each well at the final conc. of 10% (v/v). Cells were incubated for 2–4 h at 37 °C. AlamarBlue fluorescence intensity was detected by a microplate reader (Perkin Elmer, 2104 EnVision) with the excitation and emission wavelengths of 540 nm and 595 nm, respectively. For U937/iFIP1 single clones, cells were first induced with 2 μg/mL Dox for two days before JTE-607 treatment.

## Cell competition assay

HeLa, HepG2, or U937 cells containing inducible FIP1 were seeded in a 12-well plate and were induced with Dox at 2 µg/mL for 2 days. Cells were then cultured with JTE-607 (1,10, or 50 µM) or with DMSO for 3 days. Cells collected in suspension were subject to flow cytometry analysis.

## APA reporter assays

Cell transfection was carried out by using Lipofectamine 3000. To compare JTE-607 response in cells with episomal or genome-integrated reporter plasmids, we transfected HeLa Tet-on and HepG2 Tet-on cells in a 24-well plate with the plasmids PB-TRE-RiG-AD or PB-TRE-RiniG-1600-TT (300 ng/well). Six hours post transfection, cells were washed with PBS and changed to fresh media containing Dox (2 µg/mL) and JTE-607 (0, 0.1, 1, or 10 µM). All cells were incubated for 24 h post Dox induction and JTE-607 treatment. Cells were collected by trypsinization for flow cytometry analysis (BD Biosciences, LSRII). Data were analyzed by using FlowJo. Only cells with positive green and red signals were used. The $\log_2(R/G)$ value was calculated for each living cell, where R and G are red and green fluorescent signals, respectively. The fraction of long isoform was calculated by using the following formula: $2^{[\log_2(R/G)^{test} - \log_2(R/G)^{ctrl}]}$, Where, $(R/G)^{test}$ is for the pRiG vector containing a pPAS and $(R/G)^{ctrl}$ is for the control pRiG vector without a pPAS. The fraction of short isoform is 1- fraction of long isoform.

## Flow cytometry analysis of γH2AX

U937/iFIP1 cells were induced with Dox for 2 days, followed by 1 µM JTE-607 treatment for 24 h. Cells were fixed with 4% paraformaldehyde for 10 min and were permeabilized with 0.5% Triton X-100 for 3 min. Cells were washed with PBS and incubated with an anti-γH2AX antibody (Cell Signaling, Cat #: 9718, RRID: AB_2118009) with the dilution of 1:200 in PBST (PBS with 0.1% Tween-20) containing 1% BSA for 1 h at room temperature. Cells were then washed 3 times with ice-cold PBST and were incubated with an Alexa Fluor 594-labeled, goat anti-rabbit antibody (2 drops/mL, ReadyProbes, Thermo Fisher Scientific, Cat #: R37117) for 1 h at room temperature in the dark. After wash with PBST, samples were analyzed by flow cytometry (BD Biosciences, LSRII) with excitation wavelengths of 488 nm (to track FIP1 expression) and 532 nm (for γH2AX expression). Data were analyzed with FlowJo.

## Cell cycle phase analysis with propidium iodide (PI) DNA staining

Cells were harvested and washed in cold PBS, and then fixed in cold 70% ethanol for at least 2 h at 4 °C. Fixed cells were transferred to room temperature, washed with PBS, gently resuspended in 300 µL PI staining buffer (PBS containing 50 µg/mL PI, 50 µg/mL DNase-free RNase A and 0.1% BSA), and incubated for 30 min at 37 °C in the dark. Samples were analyzed by flow cytometry (BD Biosciences, LSRII) with the PI channel. Data were analyzed with FlowJo and GraphPad.

## Chromatin RNA (chrRNA) extraction and analysis

chrRNA extraction was carried out as previously described[63] with some modifications. Briefly, cells were seeded in 15-cm dishes and were treated with JTE-607 when cell confluency reached 75–80%. Adherent cells were wash with 10 mL of ice-cold PBS twice, scraped from the dishes, and collected into a 10-mL tube. Suspended cells were collected into a 10-mL tube, centrifuged at $420 \times g$ at 4 °C for 5 min, and then washed with 10 mL ice-cold PBS twice. After removal of PBS, cells were re-suspended in 4 mL of ice-cold HLB + N buffer, containing 10 mM Tris- HCl (pH 7.5), 10 mM NaCl, 2.5 mM MgCl$_2$ and 0.5% (v/) NP-40, and were left on ice for 5 min. Cells were then underlaid with 1 mL of ice-cold HLB + NS buffer (10 mM Tris-HCl (pH 7.5), 10 mM NaCl, 2.5 mM MgCl2, 0.5% (v/v) NP-40 and 10% (wt/vol) sucrose), and were centrifuged at 420x g at 4 °C for 5 min. After careful removal of the supernatant, nuclear pellets were re-suspended in 125 µl of ice-cold NUN1 buffer, containing 20 mM Tris-HCl (pH 7.9), 75 mM NaCl, 0.5 mM EDTA, and 50% (v/v) glycerol, followed by addition of 1.2 mL of ice-cold NUN2 buffer, containing 20 mM HEPES-KOH (pH 7.6), 300 mM NaCl, 0.2 mM EDTA, 7.5 mM MgCl$_2$, 1% (v/v) NP-40, and 1 M urea. Samples were vortexed at the maximum speed, followed by incubation on ice for 5 min. Chromatin pellets were collected after centrifugation at $16,000 \times g$ at 4 °C for 2 min and were re-suspended with 50 µL RNase-free water. The chromatin pellet was dispersed by using a 200 µL tip before chromatin RNA was extracted with 1 mL TRIzol. chrRNA was subject to ribosomal removal by using the illumina Ribo-Zero Plus rRNA depletion kit before cDNA library construction by using the NEBNext® Ultra™ II Directional RNA Library Prep Kit for Illumina Sequencing were carried out on an Illumina HiSeq machine at Ademera Health (South Plainfield, NJ, USA). chrRNA was also used for RT-qPCR analysis for individual genes.

## RT-qPCR

Total RNA was extracted with Trizol reagents (Thermo Fisher, Cat #: 15596026) according to manufacturer's protocol. Residual genomic DNA was digested with TURBO DNase (Invitrogen). Total RNA (1 µg) was reverse transcribed with M-MLV reverse transcriptase (Promega) and an oligo(dT) primer. chrRNA was reverse transcribed with random hexamers. qPCR was performed with Hot Start Taq-based Luna qPCR master mix (NEB) on a QuantStudio 5 Real-Time PCR System (Thermo Fisher). For readthrough analysis, primer pairs were designed to target downstream region of the last PAS. Primer pairs targeting the last intron and last exon were used for pre-mRNA quantification. Primers for RT-qPCR analysis of mature RNA expression were designed to target different exons to minimize intron contamination. PCR primer sequences are shown in Supplementary Table 1. Two-tailed student's t-test was used to calculate significance of difference.

## Immunoblotting

Cells were lysed with RIPA buffer (50 mM Tris pH 8.0, 150 mM NaCl, 1% Triton X-100, 0.5% sodium deoxycholate, 1 mM EDTA, 1 mM DTT and 1 mM PMSF) on ice for 30 min, followed by centrifugation at $16,000 \times g$ at 4 °C for 15 min. The protein concentration of cell lysate was determined by using the DC Protein Assay (Bio-Rad). Protein samples were denatured by using the SDS loading buffer (50 mM Tris-HCl pH 6.8, 2% SDS, 10% Glycerol, 0.1% Bromophenol blue and 100 mM DTT) at 95 °C for 10 min. A total of 20 µg of protein per sample was used for SDS-PAGE. After protein transfer to a PVDF membrane, the membrane was incubated with 5% non-fat milk and a primary antibody (β-actin, Santa Cruz Cat #: sc-69879, 1:2000 dilution; anti-GFP, Santa Cruz Cat #: sc-9996, 1:1000 dilution; anti-FlP1, Bethyl Cat #: A301-462A, 1:2000 dilution) at 4 °C overnight. Proteins were detected by using HRP-conjugated secondary antibodies (goat anti-mouse or anti-rabbit lgG, Jackson lmmunoResearch; Cat #:115-035-062 or Cat #: 11-035-144, respectively, 1:5000 dilution) followed by incubation with chemiluminescent substrates (Bio-Rad Clarity ECL reagent).

## Compound synergy analysis

Test compounds (anti-cancer library from Selleck Chemicals LLC, Cat# L3000; details of individual compounds are shown in Supplementary Table 2) were dissolved in DMSO and dispensed into white, 384-well tissue culture treated assay plates (Greiner 781080) using the Echo 650 acoustic liquid handling system such that the final DMSO concentration was 0.2%. After the compounds were added to the assay plate, 25 µL of U937/U937/iFIP1 cells (500 cells/well) in complete media were added to each well using the MicroFlo bulk liquid dispenser (BioTeK). After 72 h incubation at 37 °C + 5% CO$_2$, 12.5 µL of CellTiterGlo (Promega) was added to the plates, and the luminescence was measured after 30 min by using the ClarioStar Plate reader (BMG lab tech). For U937/iFIP1 single clone, cells were first induced with 2 µg/mL Dox for

two days prior to drug treatment. Compound synergy was analyzed by using the SynergyFinder 3.0 program[64] or the Combination Index (CI) method based on the Bliss independence model[65].

## RNA sequencing by QuantSeq

Library preparation of total RNA was carried out by using the Lexogen 3′ mRNA-Seq QuantSeq FWD or QuantSeq-Pool kit according to manufacturer's instructions. Library preparation, quality control, and sequencing were carried out by Ademera Health (South Plainfield, NJ, USA). cDNA libraries were sequenced on an Illumina HiSeq machine (2 × 150 nt) at Ademera Health.

## Differential gene expression analysis

QuantSeq FWD or QuantSeq-Pool data were processed according to manufacturer's recommendation. Briefly, For QuantSeq FWD data, read 1 data were first trimmed by using the BBtools[66] and then mapped to the human genome (hg19) using STAR-2.7.7a[67]. The number of reads mapped to each gene was counted by using featureCounts[68]. For QuantSeq-Pool data, raw pooled data were first demultiplexed by using the idemux tool, and the read 1 data were then used. The Umi_tools package was used to remove read duplicates based on the location of the alignment and the UMI information. The number of reads mapped to each gene was calculated by using the featureCounts tool[68]. Only genes with more than five reads in a sample were used for further analysis. The read count of each gene was normalized by the total mapped reads to the genome. PseudoCount of 1 was applied to prevent infinity values in ratio calculation. The significance of expression difference was assessed by Fisher's exact test or DESeq2. All $P$ values were adjusted by the Benjamini−Hochberg (BH) method to control the false discovery rate. BH-adjusted $P$ value < 0.05 was considered significant. A fold change of 1.2 or 2 was additionally applied to select regulated genes.

## APA analysis

QuantSeq FWD reads containing at least 15A's after adapter trimming were selected and mapped to the human genome (hg19) by using bowtie2[69]. For QuantSeq-Pool data, read 2 data were used after removal of UMI and poly(T) sequences. Reads with a mapping quality score (MAPQ) < 10 were discarded. The last aligned position (LAP) of each read was compared to annotated PASs in the polyA_DB database[70], allowing ± 24 nt flexibility. Matched reads were poly(A) site-supporting (PASS) reads, which were used for further APA analysis. For 3′ UTR APA analysis, the two PASs with the highest usage levels in the 3′UTR of the last exon were compared. One was named proximal PAS (pPAS) isoform, and the other distal PAS (dPAS) isoform. For IPA analysis, the IPA isoform with the highest expression level among all IPA isoforms was compared to all isoforms using last exon PASs (TPA isoforms) combined. Relative Expression Difference (RED) was calculated as the difference in ratio (log$_2$) of isoform abundance (dPAS isoform vs. pPAS isoform) between two comparing samples. Significant APA events were those with RED > log$_2$(1.2) or <−log$_2$(1.2) and BH-adjusted $P$ < 0.05 (Fisher's exact test).

## Gene feature analysis

Gene features were based on RefSeq annotations. RNA stability score in HepG2 cells was based on ratio (log$_2$) of RNA abundance in flow-through sample to 4sU-labeled sample, as we previously generated[13]. Distance of each gene to its nearest neighbor gene (NNG) was defined by RefSeq. A linear regression model was used to examine correlation between gene features and gene expression changes. The importance of each feature was assessed by its individual $R^2$. The cumulative $R^2$ value for a feature was based on the feature and all other features having a higher individual $R^2$.

## Gene ontology analysis

Gene ontology analysis was carried out by using the GOstats package in R[71]. Fisher's exact test was used to calculate $p$ values to indicate significance of association between a gene set and a GO term. GO terms associated with more than 1,000 genes were considered too generic and were discarded. Any GO term that overlapped with a more significant term by >75% was removed to reduce redundancy.

## Analysis of chrRNA-seq data

chrRNA-seq reads were first trimmed by using the Trim Galore tool (https://github.com/FelixKrueger/TrimGalore) to remove 5′ and 3′ adapter sequences. Trimmed reads were mapped to the human genome (hg19) using STAR-2.7.7a[67]. The number of reads mapped to each gene was calculated by using the featureCounts tool[68]. Data was normalized by the total number of reads mapped to the genome. For readthrough analysis, the 4 kb region downstream of the last PAS position (annotated by the RefSeq database) of each gene was used (readthrough analysis region). Genes whose readthrough analysis region overlapped with another gene were excluded. The readthrough score (RTS) for each gene was calculated as ratio (log$_2$) of read density in the readthrough analysis region to read density of gene body. ΔRTS was used to measure change of readthrough between two samples. For metagene analysis of chrRNA-seq data, bam files from STAR alignments were processed by the compute-Matrix function from the deepTools program[72] to calculate read coverage in the 5 kb upstream region from the TSS and 5 kb downstream region from the PAS. Extreme values (top 5% and bottom 5%) were removed. The mean read coverage at each position was calculated and plotted by using R. ChrRNA-seq Overlapping Signal Score (CROSS) for each gene was calculated as ratio (log$_2$) of number of reads mapped to antisense strand to number of reads mapped to sense strand within the last 1 kb region of each gene using data from JTE-607-treated samples. ChrRNA-seq Intervening Signal Score (CRISS) for each gene was calculated as ratio (log$_2$) of read density in the intervening region between two adjacent genes with the same transcriptional direction of JTE-607-treated sample to that of DMSO-treated sample.

## Neighboring gene pair analysis

The distance between a gene of interest (GOI) to its nearest neighbor gene (NNG) was defined by the RefSeq database. The longest isoform was used for each gene when there were multiple isoforms. GOI and NNG gene expression levels were defined by using DMSO-treated samples. To examine the influence of gene expression between neighboring genes, gene pairs were put into a 5 × 5 table (25 bins) based on GOI and NNG expression levels. Only the gene pairs with short distance (bottom 40%) were used. The median log$_2$Ratio (JTE-607 vs. DMSO) of GOI group in each cell was represented in a heatmap. A similar strategy was used to examine GOI expression in 25 bins constructed using GOI to NNG distance and CROSS value.

## Statistical analysis

When Student's $t$-test was used to determine statistical significance between groups, the two-tailed version was used unless specified otherwise. Significance of APA changes and gene expression differences was assessed by using the Fisher's exact test or DESeq2[73]. Their $P$ values were adjusted by the Benjamini−Hochberg (BH) method to control the false discovery rate. K−S (Kolmogorov−Smirnov) test (two-sided) was used to compare data distributions of different gene set. The Wilcoxon rank sum test was used to compare gene expression changes between gene sets.

## Reporting summary

Further information on research design is available in the Nature Portfolio Reporting Summary linked to this article.

## Data availability

The data supporting the findings of this study are available from the corresponding authors upon reasonable request. Sequencing datasets generated in this study have been deposited into the GEO database with the accession number GSE218557. Source data are provided with this paper.

## Code availability

Original code and any additional information required to reanalyze the data reported in this paper are available from the corresponding author upon request.

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

## Acknowledgements

We thank Yongsheng Shi and members of BT lab for helpful discussions. This work was funded by NIH grants GM084089 and GM129069 (to B.T.), 1S10OD030245 (to J.M.S.), and P30 CA010815 (to J.M.S.).

## Author contributions

Y.C., L.W. and B.T. conceived of and designed the experiments. Y.C., Q.D, J.S., and J.C. performed the experiments. L.W. analyzed the data. Q.L. provided guidance on statistical tests. J.M.S. provided guidance on experiments. Y.C., L.W. and B.T. wrote the paper.

## Competing interests

The authors declare no competing interests.
