## [Peer Review File · Nature Communications]

Elevated pre-mRNA 3' end processing activity in cancer cells renders vulnerability to inhibition of cleavage and polyadenylationREVIEWER COMMENTS

Reviewer #1 (Remarks to the Author):

Eukaryotic mRNA precursors undergo extensive processing, including the cleavage and polyadenylation (CPA) at their 3'-end. This study reveals that CPA inhibition (CPAi) leads to global transcriptional readthrough that affects the expression of the neighboring genes in high gene density regions and the reduced use of intronic polyadenylation sites. Very interestingly, cells with higher CPA activity are more sensitive to CPAi, which suggests a potentially powerful anti-cancer therapy strategy for cancer cells with high CPA activity. The findings here are very interesting and will be a great contribution to the field, although I am a bit confused by one point listed in the major concerns. I have several comments for further improvement of the manuscript.

Major Concerns

1. The authors found that FIP1 OE in U937 cells leads to higher CPA activity and much lower IC50 for JTE-607 treatment. This is an interesting finding and is one of the most important novel points of the manuscript. However, U937MP cells are of higher CPA activity and much higher IC50 for JTE-607 treatment compared to U937 cells. This is in sharp contrast to the conclusion that cells with higher CPA activity are more vulnerable to JTE-607, which makes me confused about the above conclusion. Could the author better explain this? If this conclusion does not stand, the value of this manuscript would be compromised.
2. Related to the above point, the uncoupling of degree in gene expression disturbance through CPA and the level of suppression of cell survival is very surprising. I noticed that the expression changes were measured under treatment of 1 μ M JTE-607 which means that only 20% survived according to Fig. 5B. Could the relatively small expression changes related to CPA seen in U937 compared to U937MP be caused by dying of U937 cells but not U937MP cells? I'd suggest the authors do another set of treatment using concentration lower than IC50 for both cells.
3. There is nice correlation between expression changes upon JTE-607 treatment and Pcf11 knockdown. However, about knockdown other Cpsf complex components, such as Cpsf1, Cpsf2, Cpsf3, or Cpsf4?

Minor concerns

4. The method for significantly regulated genes "($P < 0.05$, Fisher's exact test; fold change > 1.2 ; Figs. 1B and 1C)" is different from the commonly used methods, such as DESeq2 or EdgeR, which need proper justification to convince the readers that the findings here are valid using commonly used method. In addition, the criteria of fold change 1.2 is not common, this threshold also needs proper justification.
5. This conclusion "we observed general downregulation of gene expression in both HeLa and HepG2 cells after JTE-607 treatment, i.e., more genes were downregulated than upregulated (note the upregulated and downregulated gene numbers in Figs. 1B and 1C)" is not appropriate. Since

transcriptome analysis here are based on the assumption that the cells are with the same number of transcripts before and after treatment. Therefore, I'd suggest remove this part as well as similar statements related to Figure 5D which won't affect the main findings of the manuscript.

6. The RTS score is good in quantifying the readthrough. However, how is it calculated is not sufficiently described and justified. In particular for genes with multiple PASs, how to define the gene body and downstream region?

7. Is there any direct evidence that support the collision of RNA pol II as described in Fig. 2E, experimental or from the literatures?

8. Figure legends of 1E, S1B and 5D lack the descriptions of grey columns.

9. The statement "features related to gene size, such as overall intron size, gene size, and largest intron size" is confusing. Please revise the description.

10. "By the same token, PASs in cells with higher CPA activities are more inhibited by JTE-607 than cells with lower CPA activities, as in HepG2 vs. HeLa cells (illustrated in Fig. 4I)." Shall the "Fig. 4I" here be "Fig. 4J"?

11. "Fig. 7K" is not mentioned in the main text.

12. Grammar errors such as "we specially identified genes that were did not show any readthrough reads from neighboring genes within 5 kb from their TSS or PAS (Fig. S2F)".

Reviewer #2 (Remarks to the Author):

This manuscript does a nice job describing the full range of changes in polyadenylation patterns that are observed upon treating multiple human cell lines with the CPSF-73 inhibitor JTE-607 and, more interestingly, provides important mechanistic insights into why different degrees of changes are observed across cell lines. The authors find that more frequently used PASs are impacted to a greater extent by JTE-607 and that cells with elevated cleavage/polyadenylation activity, e.g. due to high FIP1 levels, display greater transcriptional readthrough and gene expression changes. A nice combination of transcriptomics, linear regression analysis, and reporter plasmids are used to support the conclusions. The writing is clear and accessible. The study overall provides important insights into how this cancer relevant compound works.

Minor points:

- Please comment on why FIP1 OE had no effect in HeLa cells (Fig 7D).
- The data in Fig 5 are intriguing as they show the degree of gene expression changes caused by JTE-607 can be uncoupled from the level of suppression of cell survival. Is there anything notable in the sets of

genes with altered expression that are different between the U937MP/U937 cell lines that might help explain this result?

Point-by-point response to reviewers' comments.

We thank the two reviewers for their enthusiasm about our work and their constructive comments! We are particularly grateful for their question about the differences between U937 and U937MP cells with respect to transcriptomic changes vs. cell death in response to JTE-607. By addressing this issue, we have now uncovered cell proliferation as another key determinant leading to JTE-607-mediated cell death. We hope our response and revision are now satisfactory to the reviewers. Our point-by-point response to their comments is detailed below.

With gratitude,

Bin Tian, PhD
The Wistar Institute

==

Reviewer #1 (Remarks to the Author):

Eukaryotic mRNA precursors undergo extensive processing, including the cleavage and polyadenylation (CPA) at their 3'-end. This study reveals that CPA inhibition (CPAi) leads to global transcriptional readthrough that affects the expression of the neighboring genes in high gene density regions and the reduced use of intronic polyadenylation sites. Very interestingly, cells with higher CPA activity are more sensitive to CPAi, which suggest a potentially powerful anti-cancer therapy strategy for cancer cells with high CPA activity. The findings here are very interesting and will be a great contribution to the field, although I am a bit confused by one point listed in the major concerns. I have several comments for further improvement of the manuscript.

Thanks for the accurate summary of our work and enthusiasm about the significance of our findings.

Major Concerns

1. The authors found that FIP1 OE in U937 cells leads to higher CPA activity and much lower IC50 for JTE-607 treatment. This is an interesting finding and is one of the most important novel points of the manuscript. However, U937MP cells are of higher CPA activity and much higher IC50 for JTE-607 treatment compared to U937 cells. This is in sharp contrast to the conclusion that cells with higher CPA activity are more vulnerable to JTE-607, which makes me confused about the above conclusion. Could the author better explain this? If this conclusion does not stand, the value of this manuscript would be compromised.

Thanks for bringing up this very important issue, which was also mentioned by reviewer 2. We have made substantial efforts to address this issue in our revision:

- 1. We now show that cell proliferation is a key factor causing cell death after JTE-607 treatment. This is based on cell cycle phase analysis of JTE-607-treated U937 cells with or without FIP1 induction (Figs. 10c-e and Supplementary Figs. 10). This view is further bolstered by the synergy analysis between JTE-607 and compounds that inhibit DNA repair pathways and DNA synthesis (Figs. 10f-h and Supplementary Figs. 11 and 12).**

2. Because U937MP cells are largely quiescent (Fig. 7b and Supplementary Figs. 7a) but also have a relatively low IC_{50} compared to other cells, such as HeLa, we conclude that transcriptomic disturbance alone can also cause JTE-607-elicited cell death, albeit less potent than DNA damage (summarized in Fig. 10i).

2. Related to the above point, the uncoupling of degree in gene expression disturbance through CPA and the level of suppression of cell survival is very surprising. I noticed that the expression changes were measured under treatment of 1 μ M JTE-607 which means that only 20% survived according to Fig. 5B. Could the relatively small expression changes related to CPA seen in U937 compared to U937MP caused by dying of U937 cells but not U937MP cells? I'd suggest the authors do another set of treatment using concentration lower than IC_{50} for both cells.

Thanks for raising this important point. We apologize for not making it clear that IC_{50} was carried out over a course of three days, whereas RNA-seq analysis was done after 7 hours of drug treatment. At the 7th hour time point, there was no sign of cell death, even with a higher concentration (10 μ M). We now show this data in Supplementary Fig. 6a. In addition, the gene expression changes in U937 and U937MP cells after 7 hours of JTE-607 treatment are well correlated with those in HepG2 cells (Supplementary Fig. 6b), indicating the transcriptomic responses are largely similar at this time point.

3. There is nice correlation between expression changes upon JTE-607 treatment and Pcf11 knockdown. However about knockdown other Cpsf complex components, such as Cpsf1, Cpsf2, Cpsf3, or Cpsf4?

Thanks for bringing up this great thought! There have been several studies so far which carried out systematic knockdown of CPA factors and examined gene expression and APA, including our previous work (Li et al. PLoS Genetics, 2015, PMC4407891). One major conclusion from these studies is that different CPA factors impact different genes and APA events differently. However, a big caveat of these studies is that they are all based on siRNA knockdown over a relatively long period of time (more than 2 days at least), which could cause many secondary effects. In addition, feedback compensatory mechanisms could also mitigate original effects. For example, in our JTE-607-treated cells, the intronic polyadenylation of PCF11 was inhibited and full-length transcripts are increased. We therefore do not think it would be fruitful to pursue a systemic comparison of our current data based on transient inhibition of CPA with previous data. We now discuss this in our revision.

Minor concerns

4. The method for significantly regulated genes "($P < 0.05$, Fisher's exact test; fold change > 1.2 ; Figs. 1B and 1C)" is different from the commonly used methods, such as DESeq2 or EdgeR, which need proper justification to convince the readers that the findings here are valid using commonly used method. In addition, the criteria of fold change 1.2 is not common, this threshold also need proper justification.

We did not use the traditional cutoffs mainly because we focused on dose-dependent events. Because we had one sample for each concentration, DESeq2 or EdgeR are not applicable. We did, however, apply the Benjamin-Hochberg method to control the false discovery rate, which we now explicitly indicate. Our choice of fold change of 1.2 is meant to identify mildly regulated genes. We have now included results using 2-fold as a cutoff (Supplementary Fig. 1a). The trend of more genes being downregulated was in fact

even stronger with fold change ≥ 2 as the cutoff. It is also noteworthy that we compared multiple cell types and they show similar gene expression changes (Supplementary Figs. 1b and 6b), further supporting the robustness of our conclusions and indicating a common CPAi transcriptome change signature.

5. This conclusion “we observed general downregulation of gene expression in both HeLa and HepG2 cells after JTE-607 treatment, i.e., more genes were downregulated than upregulated (note the upregulated and downregulated gene numbers in Figs. 1B and 1C)” is not appropriate. Since transcriptome analysis here are based on the assumption that the cells are with the same number of transcripts before and after treatment. Therefore, I’d suggest remove this part as well as similar statements related to Figure 5D which won’t affect the main findings of the manuscript.

Thanks for pointing out this issue! Our normalization method is indeed based on the assumption that the total amount of transcripts is the same across samples. The imbalance between number of upregulated and number of downregulated genes is quite reproducible across all samples. We agree that this assumption may not be correct in all cases. We have therefore toned down our interpretation of this finding and indicated this caveat.

6. The RTS score is good in quantifying the readthrough. However, how is it calculated is not sufficiently described and justified. In particular for genes with multiple PASs, how to define the gene body and downstream region?

We used the last PAS annotated in the RefSeq database as the beginning of readthrough region. We now make it more clear in our manuscript (see Materials and Methods).

7. Is there any direct evidence that support the collision of RNA pol II as described in Fig. 2E, experimental or from the literatures?

Thank you very much for bringing up this issue. We have taken it very seriously. We have developed a score, named CROSS, to indicate Pol II collision (Figs. 4c-e). In addition, for read-in analysis, we developed another score, named CRISS, to indicate Pol II read-in potential (Fig. 4k).

8. Figure legends of 1E, S1B and 5D lack the descriptions of grey columns.

Fixed.

9. The statement “features related to gene size, such as overall intron size, gene size, and largest intron size” is confusing. Please revise the description.

Fixed.

10. “By the same token, PASs in cells with higher CPA activities are more inhibited by JTE-607 than cells with lower CPA activities, as in HepG2 vs. HeLa cells (illustrated in Fig. 4I).” Shall the “Fig. 4I” here be “Fig. 4J”?

Fixed.

11. "Fig. 7K" is not mentioned in the main text.

Fixed.

12. Grammar errors such as "we specially identified genes that were did not show any readthrough reads from neighboring genes within 5 kb from their TSS or PAS (Fig. S2F)".

Fixed.

Reviewer #2 (Remarks to the Author):

This manuscript does a nice job describing the full range of changes in polyadenylation patterns that are observed upon treating multiple human cell lines with the CPSF-73 inhibitor JTE-607 and, more interestingly, provides important mechanistic insights into why different degrees of changes are observed across cell lines. The authors find that more frequently used PASs are impacted to a greater extent by JTE-607 and that cells with elevated cleavage/polyadenylation activity, e.g. due to high FIP1 levels, display greater transcriptional readthrough and gene expression changes. A nice combination of transcriptomics, linear regression analysis, and reporter plasmids are used to support the conclusions. The writing is clear and accessible. The study overall provides important insights into how this cancer relevant compound works.

Thanks for accurate summary of our work and enthusiasm about the significance of our findings.

Minor points:

- Please comment on why FIP1 OE had no effect in HeLa cells (Fig 7D).

Thanks for this excellent question! This is because HeLa cells tolerate JTE-607 better. We have repeated this experiment by using a higher conc. of JTE-607 (50 μ M). The result is consistent with HepG2 data using a lower conc. of JTE-607 (1 μ M). The data are in Figs. 8g-j.

- The data in Fig 5 are intriguing as they show the degree of gene expression changes caused by JTE-607 can be uncoupled from the level of suppression of cell survival. Is there anything notable in the sets of genes with altered expression that are different between the U937MP/U937 cell lines that might help explain this result?

Thanks for bringing up this very important issue, which was also mentioned by reviewer 1. We have made substantial efforts to address this issue in our revision:

- 1. We now show that cell proliferation is a key factor causing cell death after JTE-607 treatment. This is based on cell cycle phase analysis of JTE-607-treated U937 cells with or without FIP1 induction (Figs. 10c-e and Supplementary Figs. 10). This view is further bolstered by the synergy analysis between JTE-607 and compounds that inhibit DNA repair pathways and DNA synthesis (Figs. 10f-h and Supplementary Figs. 11 and 12).**
- 2. Because U937MP cells are largely quiescent (Fig. 7b and Supplementary Figs. 7a) but also have a relatively low IC_{50} compared to other cells, such as HeLa, we conclude that transcriptomic disturbance alone can also cause JTE-607-elicited cell death, albeit less potent than DNA damage (summarized in Fig. 10i).**

REVIEWERS' COMMENTS

Reviewer #1 (Remarks to the Author):

My concerns have been adequately addressed.

The finding of cell proliferation as another key determinant leading to JTE-607-mediated cell death is very nice that has cleared up my previous confusion. In addition, it is nice to see the CROSS and CRISS scores in the revised manuscript which make the manuscript clearer.